# An assessment of the species diversity and disease potential of *Pythium* communities in Europe

Wilken Boie[1,5], Markus Schemmel[1,5], Wanzhi Ye[1], Mario Hasler[2], Melanie Goll[3], Joseph-Alexander Verreet[4] & Daguang Cai [1] ✉

*Pythium* sensu lato (s.l.) is a genus of parasitic oomycetes that poses a serious threat to agricultural production worldwide, but their severity is often neglected because little knowledge about them is available. Using an internal transcribed spacer (ITS) amplicon-based-metagenomics approach, we investigate the occurrence, abundance, and diversity of *Pythium* spp. s.l. in 127 corn fields of 11 European countries from the years 2019 to 2021. We also identify 73 species, with up to 20 species in a single soil sample, and the prevalent species, which show high species diversity, varying disease potential, and are widespread in most countries. Further, we show species-species co-occurrence patterns considering all detected species and link species abundance to soil parameter using the LUCAS topsoil dataset. Infection experiments with recovered isolates show that *Pythium* s.l. differ in disease potential, and that effective interference with plant hormone networks suppressing JA (jasmonate)-mediated defenses is an essential component of the virulence mechanism of *Pythium* s.l. species. This study provides a valuable dataset that enables deep insights into the structure and species diversity of *Pythium* s.l. communities in European corn fields and knowledge for better understanding plant-Pythium interactions, facilitating the development of an effective strategy to cope with this pathogen.

*Pythium* (nom. cons.) sensu lato (s.l.) is a genus that includes many important species, of which more than 300 species are soil-borne plant pathogens[1,2]. Taxonomically, *Pythium* s.l. is classified in the domain Eukarya, the kingdom Chromista, the phylum Oomycota, the class Oomycetes, the order Pythiales and the family Pythiaceae. As an opportunistic phytopathogen, *Pythium* s.l. infects its hosts mainly in times of stress and susceptibility, such as during germination[3–6]. Molecular phylogenetic studies suggested a paraphyly of *Pythium* s.l., which lead to a proposal by Uzuhashi's group in 2010 to distinguish the genus more precisely. Based on morphological characteristics and

phylogenies of the mt-cytochrome c oxidase subunit 2 (*cox2*) and the D1-D2 domains of the nuc-28S rDNA[7], subdivided *Pythium* s.l. into five genera: *Pythium* sensu stricto (s.s.) (clades A-D), *Globisporangium* (clades E-G, I and J), *Elongisporangium* (clade H), *Ovatisporangium* (clade K, synonym *Phytopythium*) and *Pilasporangium* (differs from the 11 lettered clades). The infection can occur at various plant development stages, with infection of seeds during or before germination being the most important[8]. The wide host range of *Pythium* s.l. causes severe yield and quality losses of crops and agricultural products. Among the ten most destructive corn diseases in 2012, the genus

[1]Molecular Phytopathology and Biotechnology, Institute for Phytopathology, Christian-Albrechts-University of Kiel, Hermann-Rodewald Str. 9, Kiel, Germany. [2]Lehrfach Variationsstatistik, Christian-Albrechts-University of Kiel, Hermann-Rodewald Str. 9, Kiel, Germany. [3]Syngenta Agro GmbH, Lindleystraße 8 D, Frankfurt am Main, Germany. [4]Phytopathology and Crop Protection, Institute for Phytopathology, Christian-Albrechts-University of Kiel, Hermann-Rodewald Str. 9, Kiel, Germany. [5]These authors contributed equally: Wilken Boie, Markus Schemmel. ✉e-mail: dcai@phytomed.uni-kiel.de

*Pythium* s.l. accounted for about 11% yield loss in the northern US[9], with root rot and damping-off being the major plant diseases[10-12]. The dilemma is that in most cases, the *Pythium* s.l. infection goes unnoticed as it occurs below ground, and the symptoms are often misdiagnosed as nutrient deficiencies. The disease potential of *Pythium* s.l. is therefore greatly underestimated or completely neglected[12].

Although infection usually causes only minor damage to plants, it can lead to significant yield and quality losses[8]. Among crops, corn and soybeans are highly susceptible to *Pythium* s.l., with e.g. *G. irregulare* and *G. ultimum* being particularly infectious in the United States[3,13]. *Pythium* s.l. differ in their pathogenicity and disease potential[14]. For instance, *G. ultimum* is particularly aggressive with a broad host range, while *G. sylvaticum* is a highly pathogenic species for soybeans but has moderate virulence in corn as observed in the US[3]. Infection and damage from *G. ultimum* occur in Australia, Brazil, Canada, China, Japan, Korea, South Africa, and many others[14]. In Europe, intense corn cultivation increases the risk of diseases[15], such as damping off[4] and root rot[16-18]. Several pathogenic species of *Pythium* s.l. affect the growth of corn seedlings at varying initial temperatures[5,19]. However, there is currently very limited information about the disease potential, the occurrence, distribution, and species diversity of *Pythium* s.l. communities in Europe.

Meta-barcoding, aided by next-generation sequencing (NGS), revolutionizes microbiome survey research, enabling the comprehensive identification of different organisms, or species even within an environment. The most commonly used genomic markers in soil microbiome research are the Internal Transcribed Spacer (ITS) region and the 16S-rRNA marker. Both genetic regions contain highly conserved and variable regions, allowing the development of primers that bind to the conserved regions, while the variable regions in between are used for taxonomic determination and phylogenetic analysis[20-23]. In addition to ITS region[24-26], the Cytochrome c oxidase subunit 1 (CO1, COX1) is frequently used for the taxonomic identification of oomycetes[27,28] and *Pythium* spp. s.l.[29] Also, the divergent regions D1-D3 of the large subunit (LSU) of ribosomal DNA are suitable for the taxonomic identification of oomycetes[25,30]. Nevertheless, due to the highly conserved nature of LSU, some closely related species could not be taxonomically differentiated using LSU, and also ITS and CO1 often fails to distinguish certain species because of barcoding gaps[25].

Efforts are underway worldwide to identify host resistance and genes against *Pythium* spp. s.l.[31,32]. Recently, more than 60 potential resistance loci against corn stalk rot, caused by *P. aristosporum*, were reported from a genome-wide association study in corn[33]. However, the high species diversity of *Pythium* s.l. communities and the complex pathogenicity of the different species hamper resistance breeding[32]. No effective resistance genes and hybrids are available for corn[34], and also, the molecular interactions between corn and *Pythium* s.l. are not fully understood. Transcriptional regulation is a critical process in plant response to pathogen attacks, often controlled by hormone signaling networks, in which SA and JA are key players[35,36]. Both JA biosynthesis and JA-controlled resistance responses are essential for corn immunity to *Pythium* s.l.[37]. In addition to transcriptional regulation, miRNAs form an additional regulatory layer controlling the defense response to *G. ultimum* infection in a post-transcriptional manner[38]. A deep understanding of the molecular mechanisms underlying the plant-Pythium interactions may help develop novel strategies to manage *Pythium* s.l.

In this study, we investigate the occurrence, abundance, and species diversity of *Pythium* s.l. in corn fields across Europe using a meta-barcoding approach. Over a period from 2019–2021, soil samples from 127 sites across 11 countries were collected for this study. In total, we identify 73 species of *Pythium* s.l. including *Pythium* s.s., *Globisporangium* and *Ovatisporangium* (*Phytopythium*), with soil samples containing up to 20 species. We identify and characterize several species prevalent in European corn fields, showing high species diversity and differences in their disease potential. We demonstrate that an effective interference with plant hormone networks to suppress JA (jasmonate)-mediated defenses may be a crucial part of the virulence mechanism of pathogenic *Pythium* spp. s.l.

## Results

### *Pythium* s.l. communities are highly diverse across Europe

A set of 127 soil samples collected from France, Italy, Germany, Belgium, the Netherlands, Spain, Romania, Hungary, the Czech Republic, Switzerland, and Austria (Fig. 1) were deployed for ITS amplicon sequencing. ITS-sequence analysis revealed a high diversity of *Pythium* s.l. (Fig. 2) in soil samples across Europe. In total, 73 different species of *Pythium* s.l. were detected, from which 3 were annotated as *Phytopythium*, 30 as *Pythium* s.s. and 40 as *Globisporangium*. Noticeably, a single soil sample (P20_L27) contained up to 20 species, while no *Pythium* s.l. was detectable in two samples (P20_L07 and P20_L37, Fig. 2). As illustrated by the heatmaps and the annotation of the data from three sampling years (Fig. 2), multiple species were profoundly dominant; 46 different species were detected in 2019, from which the ten most common species were *G. attrantheridium* (33), *G. heterothallicum* (31), *G. sylvaticum* (31), *P. monospermum* (30), *P.* aff. *hydnosporum* (25), *P. arrhenomanes* (21), *G. ultimum* var. *ultimum* (21), *G. rostratifingens* (20), *G. intermedium* (18) and *G. apiculatum* (12). In 2019, the highest diversity was observed at site P19_L15 with 15 different species of *Pythium* s.l., while the lowest was at site P19_43 with 3 species. Similar results were given for 2020 and 2021 (Fig. 2). Although sampling sites decreased to 38 locations in 2020, the number of species increased to 56. The most frequent species include *G. attrantheridium* (27), *P.* aff. *hydnosporum* (26),

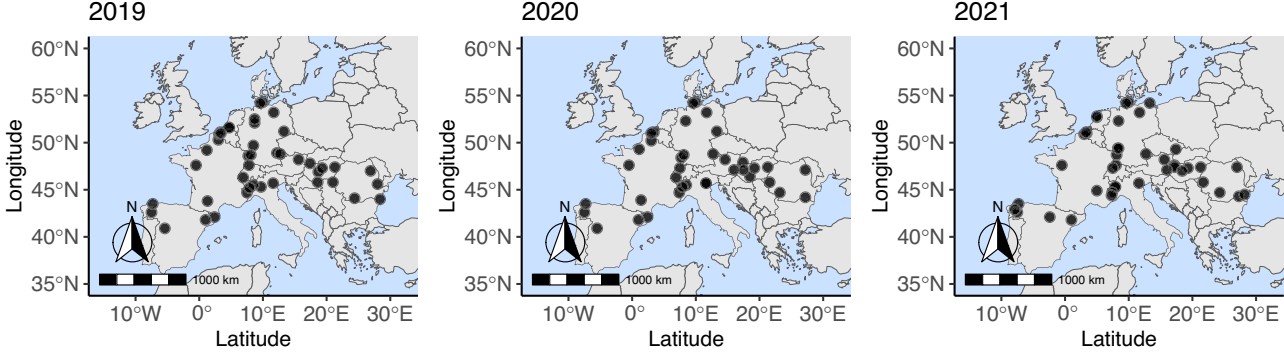

**Fig. 1 | Overview of all sampled locations across Europe.** Geographic distribution of all 127 sampled locations from 11 countries including France, Italy, Germany, Belgium, Netherlands, Spain, Romania, Hungary, Austria, Switzerland and Czech Republic. Sampling was conducted at 45 sites in 10 countries in 2019, 38 in 9 countries in 2020, and 44 in 11 countries in 2021.

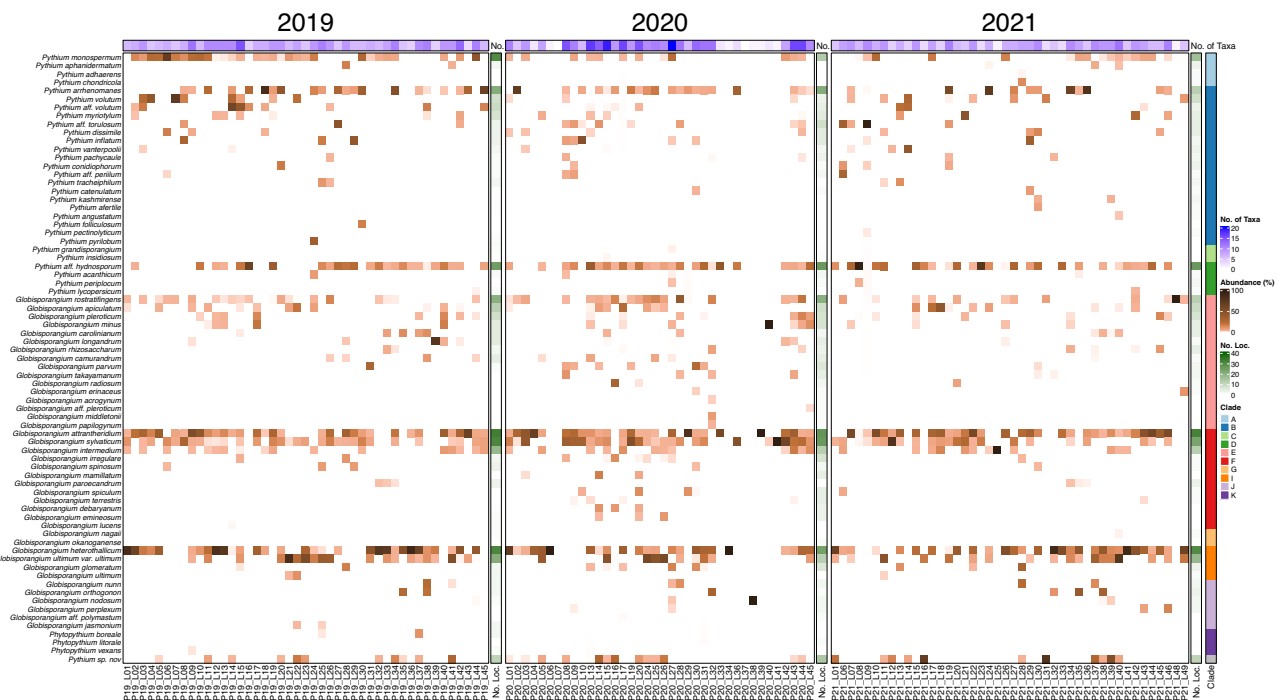

**Fig. 2 | Relative abundance of *Pythium* s.l. across all three years.** Heatmaps of relative *Pythium* s.l. abundance of sampled locations from 2019, 2020 and 2021 with the corresponding clades, and pooled for all samples in a sampling year.

*G. sylvaticum* (26), *G. heterothallicum* (24), *P. arrhenomanes* (21), *G. rostratifingens* (20), *P. monospermum* (13), *G. intermedium* (12), *G. ultimum* var. *ultimum* (12) and *G. apiculatum* (8). In 2021, the high species diversity of 57 *Pythium* s.l. species were detected at 44 sampled locations, represented primarily by *G. attrantheridium* (32), *G. heterothallicum* (30), *P.* aff. *hydnosporum* (26), *G. sylvaticum* (23), *G. ultimum* var. *ultimum* (18), *P. monospermum* (16), *P. arrhenomanes* (12), *G. intermedium* (12), *G. rostratifingens* (12) and *G. apiculatum* (9). The highest diversity was detected at site P21_L19 with 12 species, whereas P21_L48 had the lowest diversity with only one detected species (Fig. 2). Several species were frequently found at many sampled sites, which are *G. attrantheridium* (92 sites), *G. heterothallicum* (85), *G. sylvaticum* (80), and *P.* aff. *hydnosporum* (77), respectively. Six other species were identified at different sites (*P. monospermum* at 59 sites, *P. arrhenomanes* at 54 sites, *G. rostratifingens* at 52 sites, *G. ultimum* var. *ultimum* at 51 sites, *G. intermedium* at 42 sites, and *G. apiculatum*. at 29 sites), these constitute the ten most abundant *Pythium* s.l. species in all European soils investigated.

The use of bait samples allows an additional assessment of *Pythium* s.l. distribution in Europe. Fewer species were detected from the bait samples than in the soil samples (Fig. 3). Of the ten predominant species in the soil samples, eight can also be classified as predominant in the bait samples, which underlines their dominant role in Europe. They are *G. sylvaticum* (76 sites), *G. attrantheridium* (66), *G. ultimum* var. *ultimum* (56), *G. heterothallicum* (54), *G. intermedium* (40), *G. apiculatum* (33), *P.* aff. *hydnosporum* (23), and *G. rostratifingens* (17) were very common at most sampled sites (Fig. 3). In addition, the two species *P.* aff. *torulosum* (22) and *Phytopythium boreale* (15) were among the ten most abundant species in the bait samples.

### The relative abundance of *Pythium* s.l. is similar in all years
The relative abundance of the ten most prevalent species of *Pythium* s.l. identified in each sampling year were calculated. As shown in the bar- and boxplots (Fig. 4), the occurrence and abundance of these were, for the most part, consistent over the years. The abundance of the ten species was similar yearly, whereas the abundance of *G. sylvaticum* and

*P. monospermum* varied considerably among the three sampling years. In 2020, *G. sylvaticum* was more abundant than in 2019 and 2021. Furthermore, the corresponding boxplot illustrates that relative abundance variation was greater in 2020 than in the other two years. In contrast *P. monospermum* was more abundant in 2019 compared with 2020 and 2021. These data indicate that the relative abundance of *Pythium* s.l. fluctuated to some extent over the three years, but significant changes were observed only for *G. sylvaticum* and *P. monospermum*.

### The occurrence and frequency of *Pythium* s.l. varied in European countries
To assess the occurrence and abundance of *Pythium* s.l. in European countries, we compared the relative abundance of the ten most common species with the grand mean of all countries. As shown in Fig. 5A and Table S7, G. *ultimum* var. *ultimum* was significantly ($p < 0.05$) more abundant in Belgium and tended ($p < 0.1$) to be more abundant in the Netherlands. While *G. heterothallicum* is significantly more abundant in France and Romania compared with the mean of all countries, *G. apiculatum* tended to be more abundant in Germany and *P. monospermum* more in Italy. There are very few statistically significant differences between the various countries. The diversity of all *Pythium* s.l. was additionally investigated under consideration of the influencing factor of soil texture class. Depending on their clay content (Table S8), the investigated soils were divided into three classes: light (<15%), medium (15–25%) and heavy (>25%). We defined *Pythium* s.l. density by the total number of reads detected in the soil sample of a respective site. Interestingly, the light soils showed the highest *Pythium* s.l. density, followed by the medium and heavy soils. None of these soil texture classes differ significantly from the mean value of all classes (Table S9). In addition to the lower inoculum density, a lower dispersion of *Pythium* s.l. abundance in the heavy soils was observed, while the highest dispersion occurred in the light soils (Fig. 5B). Furthermore, the species diversity was examined to assess the impacts of the soils. The highest diversity of different species was found in the medium soils and differed tendentially ($p = 0.08903$) from the mean of all soil texture classes (Table S10), while the light soils, even with a

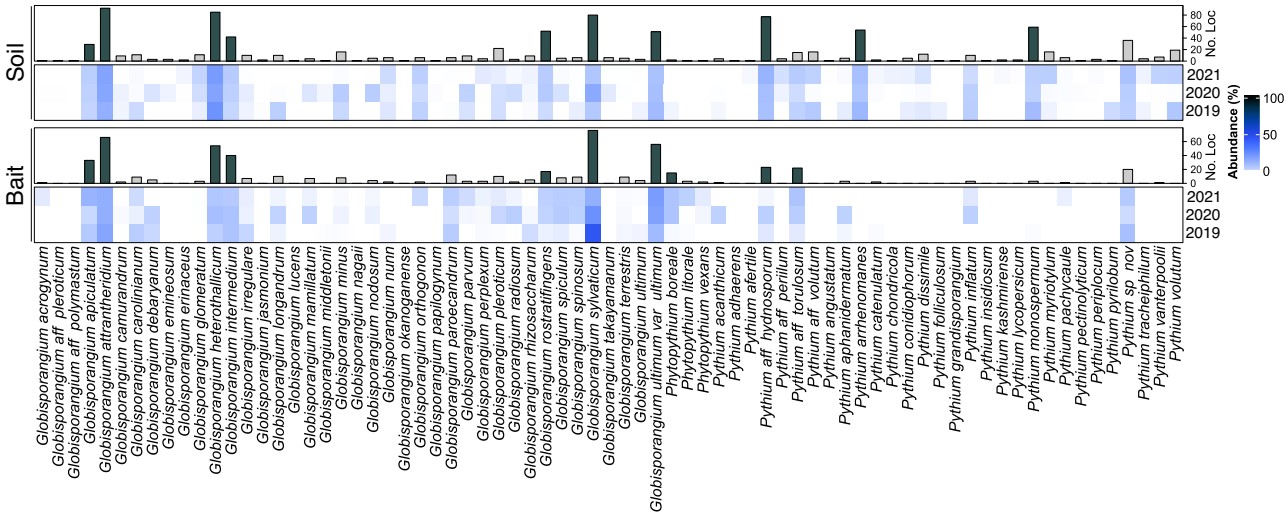

**Fig. 3 | Comparison of *Pythium* s.l. abundance of soil and bait samples.** Heatmaps of relative Pythium s.l. abundance in soil and bait samples from 2019, 2020 and 2021. The bar annotation shows the number of locations of the respective species in all three years.

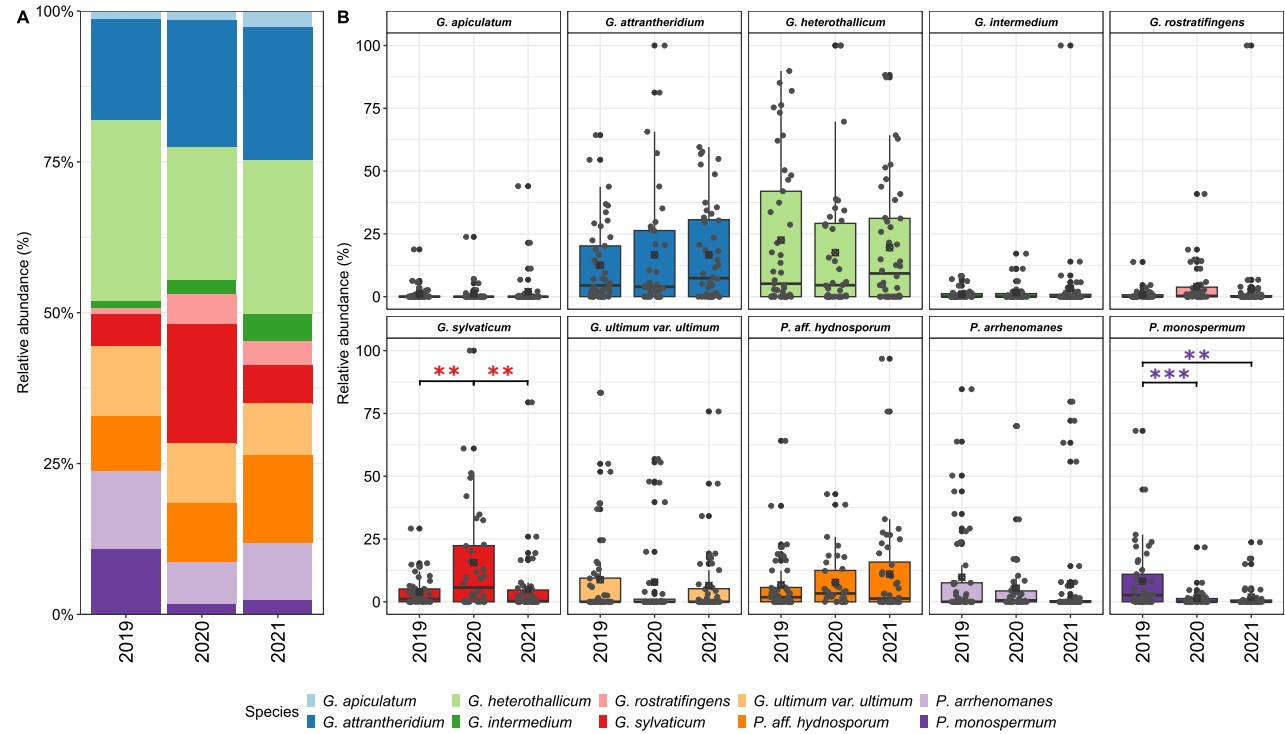

**Fig. 4 | Relative abundance of the top 10 most abundant species of *Pythium* s.l.** Barplot (**A**) and Boxplot (**B**) of relative abundance pooled for the three sampling years 2019, 2020, 2021 of the ten most abundant *Pythium* s.l. species *G. apiculatum*, *G. attrantheridium*, *G. heterothallicum*, *G. intermedium*, *G. rostratifingens*, *G. sylvaticum* and *G. ultimum* var. *ultimum*, *P. aff. hydnosporum*, *P. arrhenomanes* and

*P. monospermum*. Crossed circles represent the mean values of relative abundance of the respective species. Significant differences were tested two-sided against the mean (grand mean) of all samples for ***$p < 0.001$, **$p < 0.01$, *$p < 0.05$ (degrees of freedom = 112), $N = 127$.

similar median level as the medium soils, did not show any difference from the mean. For the heavier soils, this difference is lower than the mean of all soil texture classes but not significant (Fig. 5C).

## ASV-based diversity varies greatly between species of *Pythium* s.l

The ASVs were used to evaluate the diversity of *Pythium* s.l. between years as well as among *Pythium* s.l. species. Figure 6 summarizes the numbers of different ASVs identified in each of the three individual sample years and in all three combined years. For this purpose, the

sequenced ITS region and the resulting ASVs were used to compare the abundance of the ten most abundant species of *Pythium* s.l., *P.* aff. *hydnosporum*, *G. attrantheridium*, *G. rostratifingens*, *P.* sp. *nov*, *G. sylvaticum*, and *G. ultimum* var. *ultimum* showed a higher diversity in 2020 than in 2019 and 2021 (Fig. 6), while the ASV counts for *P. arrhenomanes* and *G. intermedium* were same in 2019 and 2020. The diversity of ASVs for *G. heterothallicum and P. monospermum* was the highest in 2019, from which G. *heterothallicum* had the highest ASV count in 2019, while *P. monospermum* remains stable between years. It was noticeable that *G. heterothallicum* had the highest diversity in all

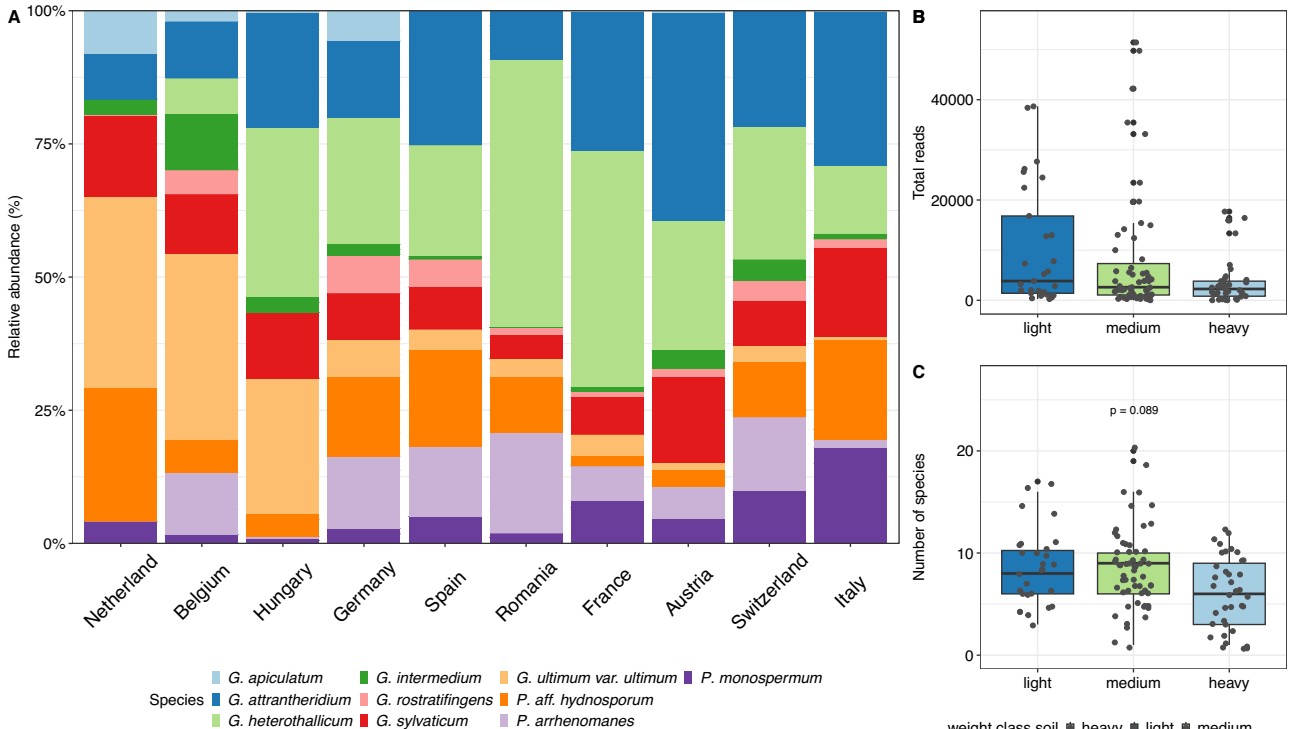

**Fig. 5 | Abundance of *Pythium* s.l. by geographic distribution and soil class.** Relative abundance of the ten most abundant *Pythium* s.l. species *G. apiculatum, G. attrantheridium, G. heterothallicum, G. intermedium, G. rostratifingens, G. sylvaticum* and *G. ultimum* var. *ultimum, P. aff. hydnosporum, P. arrhenomanes* and *P. monospermum*, pooled for the sampled countries Netherlands, Belgium, Hungary, Germany, Spain, Romania, Switzerland, France, Austria and Italy (Czech Republic excluded). Significant differences were tested two-sided against the mean of all countries (grand mean) for \*\*\**p* < 0.001, \*\**p* < 0.01, \**p* < 0.05 (degrees of freedom = 112), *N* = 126 (**A**). *Pythium* s.l. abundance of the sites classified into light, medium and heavy soil texture compared to the mean of *Pythium* s.l. abundance of all sites (grand mean). Significant differences were tested two-sided against the mean of all samples for \*\*\**p* < 0.001, \*\**p* < 0.01, \**p* < 0.05 (degrees of freedom = 110), *N* = 127 (**B**). Number of different species of *Pythium* s.l. for sites with the three soil texture classes light, medium and heavy compared to the mean of all sites (grand mean). Significant differences were tested two-sided for \*\*\**p* < 0.001, \*\**p* < 0.01, \**p* < 0.05 (degrees of freedom = 110), *N* = 127 (**C**).

three years, followed by *G. attrantheridium, P. monospermum*, and *G. sylvaticum*. Similarly, most ASVs for *G. sylvaticum* were identified in 2020 (16) and only 4 in 2019 and 2 in 2021, respectively.

Similarly, 13 ASVs were taxonomically assigned to *G. rostratifingens* in 2020, and only 3 in 2019 and 2021. The same trend was observed for *G. ultimum* var. *ultimum*, although the differences were slightly smaller with 7 ASVs in 2020 and 4 and 2 ASVs in 2019 and 2021. While *P. arrhenomanes* showed a higher ASV count in 2019 and 2020 than in 2021, in contrast to *P. aff. hydnosporum*, which exhibits a lower ASV count in 2019 and 2020. A total of 19 and 18 ASVs were taxonomically assigned to *G. sylvaticum* and *P. monospermum*. We identified 22 ASVs for *P.* sp. *nov*, 15 for *G. rostratifingens*, 13 for *G. intermedium*, and 8 for *G. ultimum* var. *ultimum*. Therefore, the lowest ASV number was observed for *G. ultimum* var. *ultimum* (Fig. 6A). In addition to the number of different ASVs, we analyzed their corresponding occurrence.

Figure 6B shows the number of locations for the ten most abundant ASVs of the ten most abundant species of *Pythium* s.l. For *P.* aff. *hydnosporum, G. apiculatum, G. attrantheridium, G. intermedium, G. rostratifingens, G. sylvaticum*, and *G. ultimum* var. *ultimum*, only one ASV was detected at each of almost all locations where the respective species was detected. This was for *P.* aff. *hydnosporum* at 75 of 77, for *G. apiculatum* at 25 of 36, for *G. attrantheridium* at 88 of 92, for *G. intermedium* at 28 of 42, for *G. rostratifingens* at 46 of 50, for *G. sylvaticum* at 79 of 80 and for *G. ultimum* var. *ultimum* at 50 of 51 locations, respectively. This effect was particularly pronounced for *P.* aff. *hydnosporum, G. sylvaticum, G. rostratifingens* and *G. ultimum* var. *ultimum*. The dominance of a single ASV is also evident for

*G. apiculatum, G. attrantheridium*, and *G. intermedium*, although this effect is somehow less pronounced for these species. Notably, only 7 and 8 different ASVs were identified for *G. apiculatum* and *G. ultimum* var. *ultimum*, respectively. It is also important to consider that *G. apiculatum* and *G. intermedium* were detected at fewer locations overall compared to species such as *G. attrantheridium* or *G. sylvaticum*. However, for other species, such as *P. arrhenomanes, G. heterothallicum*, and *P. monospermum*, this was not the case. Multiple ASVs were frequently detected for each of these three species.

## Co-occurrence between *Pythium* s.l. and their linkage with soil parameters

To investigate co-occurrence patterns between *Pythium* s.l. species, Pearson correlations were calculated (Fig. 7A). Overall, negative correlations were predominant for all 73 identified species (1853 negative, 848 positive). While *G. camurandrum* showed the highest number of significant correlations (73), *P. acanthium* exhibits the lowest (Fig. 7B). The highest number of negative correlations was observed for *G. myriotylum*. Interestingly, the top ten most abundant *Pythium* s.l. showed predominantly positive correlations. The strongest negative correlation was given between *P. monospermum* and *G. ultimum* var. *ultimum*.

Only a fraction of *Pythium* s.l. species showed a significant correlation with the pH value and the organic content of the soil. *G. debaryanum* is solely negatively correlated with soil pH. Regarding the organic matter content in the soil, positive correlations were found to be related to *P. inflatum* and *P.* sp. *nov*. (Fig. 8).

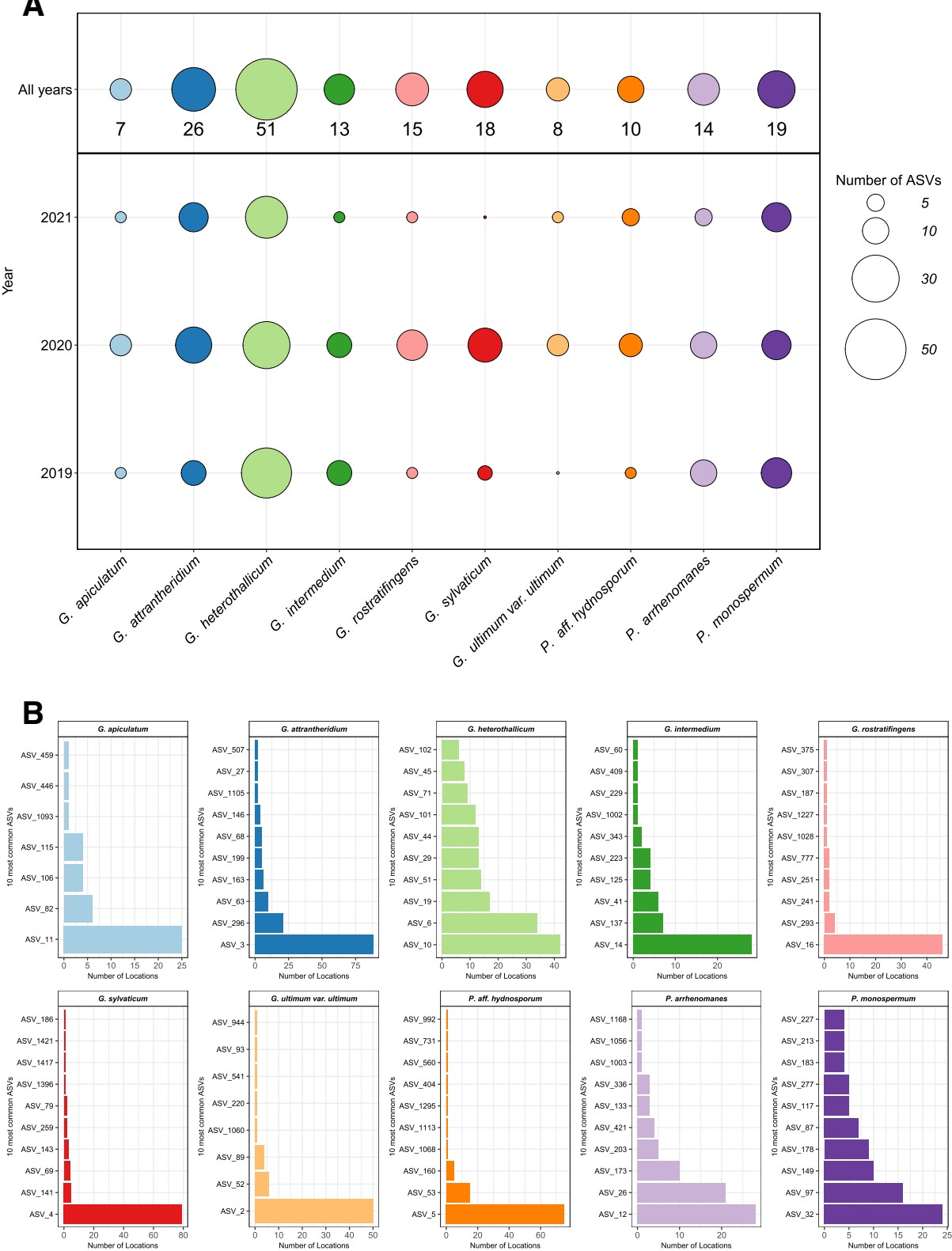

**Fig. 6 | ASV-based diversity analysis of *Pythium* s.l.** Number of amplicon sequence variants (ASVs) of the ten most abundant species of *Pythium* s.l. (*G. apiculatum*, *G. attrantheridium*, *G. heterothallicum*, *G. intermedium*, *G. rostratifingens*, *G. sylvaticum* and *G. ultimum* var. *ultimum*, *P. aff. hydnosporum*, *P. arrhenomanes* and *P. monospermum*) for the years 2019, 2020, 2021, separately and pooled (**A**). Number of locations of the ten most abundant amplicon sequence variants (ASVs) of the ten most abundant species of *Pythium* s.l. (*P. aff. hydnosporum*, *G. apiculatum*, *P. arrhenomanes*, *G. attrantheridium*, *G. heterothallicum*, *G. intermedium*, *P. monospermum*, *G. rostratifingens*, *G. sylvaticum* and *G. ultimum* var. *ultimum*) (**B**).

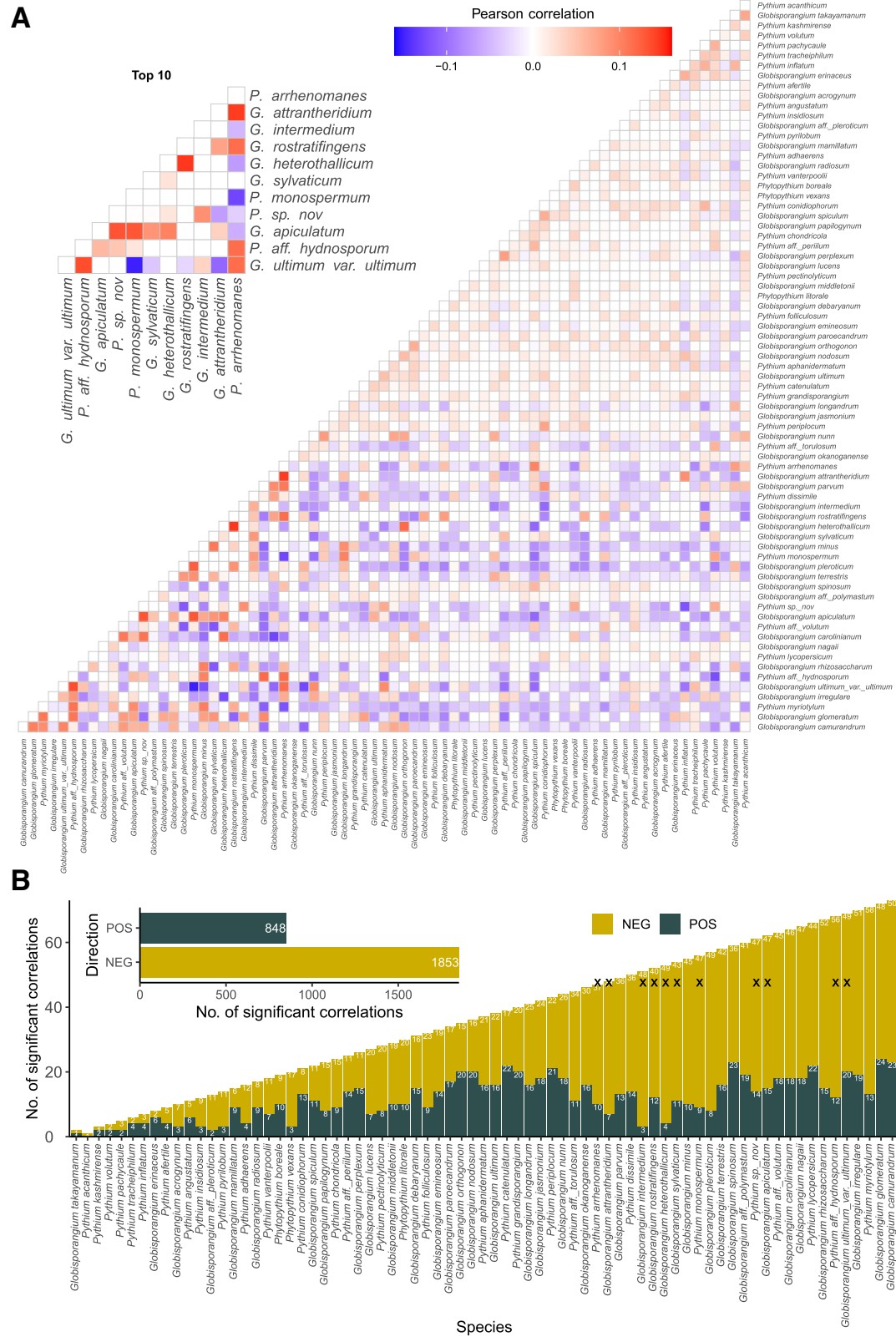

**Fig. 7 | Co-occurrence analysis of *Pythium* s.l.** Number of co-occurrence patterns (pearson correlation) of all identified species of *Pythium* s.l. (**A**) and the ratio of the direction of the correlation per species (**B**). Black crosses indicate the top 10 most abundant species. NEG = negative, POS = positive.

## *Pythium* s.l. differ in their disease potential on corn

In order to evaluate the pathogenicity of the *Pythium* s.l. species identified in this study, we conducted infection experiments on corn using *G. ultimum* var. *ultimum* and *G. attrantheridium*, and compared the germination rate, shoot fresh, and root dry weight of infected plants with those of non-infected plants. The results showed that both species caused characteristic disease symptoms but differed in severity. Thereby, *G. ultimum* var. *ultimum* showed more pronounced disease symptoms than *G. attrantheridium* (Fig. 9). The germination rate of corn was severely affected by infection with *G. ultimum* var.

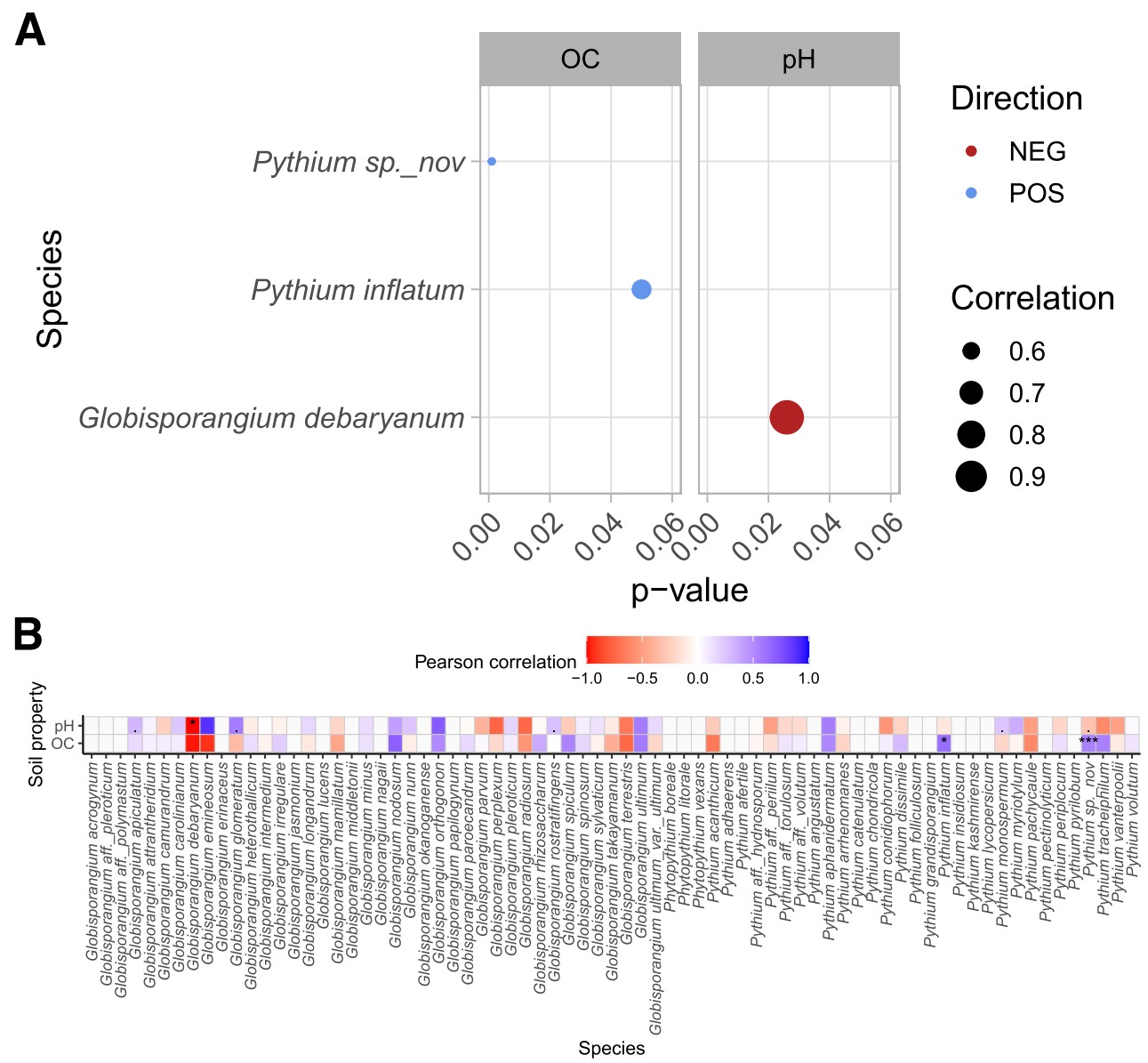

**Fig. 8 | Linkage analysis of *Pythium* s.l. with soil parameters.** Pearson correlation (two-sided) of soil pH (measured in CaCl2) and organic content (OC) of significantly (***$p < 0.001$, **$p < 0.01$, *$p < 0.05$) correlated species across Europe (**A**). Overview of *Pythium* s.l. linked to soil pH and soil organic content (**B**). OC = organic content, NEG = negative correlation, POS = positive correlation.

*ultimum*, but not with *G. attrantheridium* (Fig. 9A, Table S11). Furthermore, a reduction in shoot fresh- and root dry weight was caused by *G. ultimum* var. *ultimum*, although this reduction also occurred in *G. attrantheridium* (Fig. 9A, D, E, F, G, Table S12, and Table S13). In order to understand the underlying mechanisms, we monitored plant response by comparing the expression levels of 4 phytohormone-marker genes involved in plant defense response using RT-qPCR (Fig. 10). They were *PR1* (SA marker), *PDF1.2* (JA marker), and *ETR2* (ET marker) as well as *NCED3* (ABA marker). Remarkably, we found that the expression of *PR1* was high in roots infected with either *G. ultimum* var. *ultimum* or *G. attrantheridium* compared with the non-infected control. While the expression levels of *ETR2* and *NCED3* did not alternate substantially, *PDF1.2*, a marker gene for JA-mediated defense response, was significantly suppressed in roots infected by *G. ultimum* var. *ultimum*, but not by *G. attrantheridium* (Fig. 10, Table S14).

## Discussion

### NGS-based identification of *Pythium* s.l. in Europe

A total of 73 *Pythium* s.l. including 3 *Phytopythium*, 30 *Pythium* s.s. and 40 *Globisporangium* species, were detected in 127 soil samples. Except for two sample sites (P20_L07 and P20_L37), *Pythium* s.l. was detected at every sampled site in this study. Although the number of species in individual soil samples varied considerably, up to 20 species were found at one sampling site. This is in line with the report[3] that 11 *Pythium* s.l. species were detected from 42 production fields in Ohio and 27 *Pythium* s.l. species from 12 sample sites using ITS primers UN-UP18S42 and UNLO28S22[39]. Similarly, 51 *Pythium* s.l. species were identified from 64 fields in 2011 and 54 from 61 fields in 2012 from soybean roots by Sanger sequencing-based indentification[40]. A high number of different *Pythium* s.l. species from 11 sampling sites in Michigan, USA, were detected by sequencing the *cox1* oomycete region[41]. The higher number of different *Pythium* s.l. species in our study could reflect that the samples were taken from sites that differ

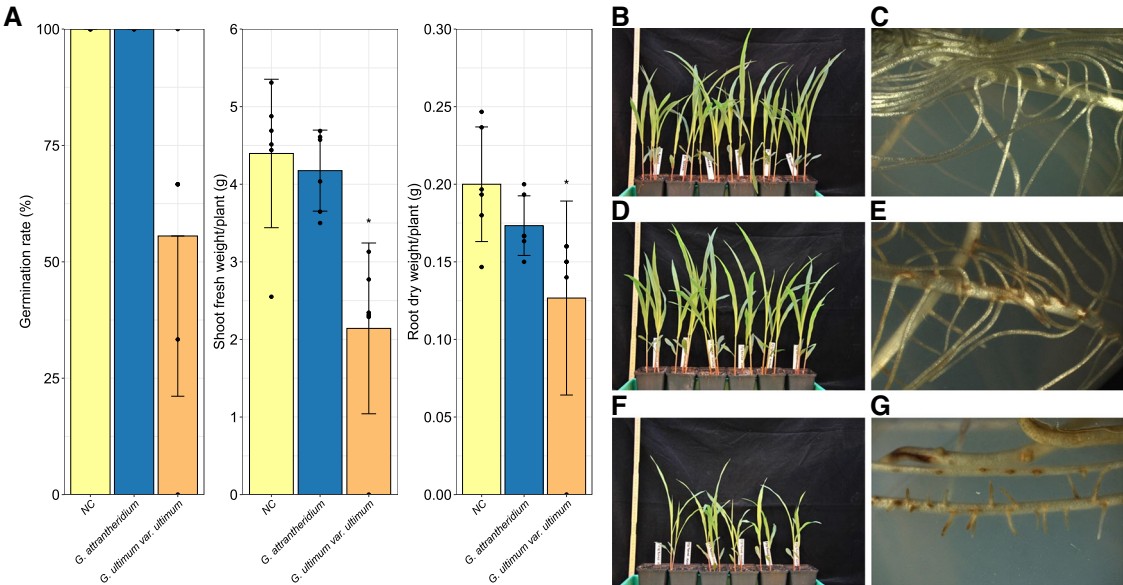

**Fig. 9 | Effects of *Pythium* s.l. infection on plant growth.** Influence of *G. attrantheridium* and *G. ultimum* var. *ultimum* on corn after infection. Comparison of germination capacity, mean shoot fresh weight, and mean root dry weight of corn plants infected with *G. attrantheridium* or *G. ultimum* var. *ultimum* (**A**), compared to the non-infected control (NC). Statistical differences were tested two-sided by a multiple contrast test for ***$p < 0.001$, **$p < 0.01$, *$p < 0.05$ (degrees of freedom = 110). Error bars indicate the standard deviation ($n = 6$). Non-infected corn plant (**B**). Stereomicroscopic image of non-infected corn roots in ½ MS medium (**C**). Corn plants infected with *G. attrantheridium* (**D**). Stereomicroscopic image of corn roots infected with *G. attrantheridium* in ½ MS medium (**E**). Corn plants infected with *G. ultimum* var. *ultimum* (**F**). Stereomicroscopic image of corn roots infected with *G. ultimum* var. *ultimum* in ½ MS medium (**G**).

greatly from each other in terms of climatic conditions and agronomic cultivation parameters. Compared with individual studies from the US, the diversity in Europe is even higher.

Taxonomic resolution depends on the used primers and the sequenced marker region. Application of *Pythium* s.l. specific primers greatly increased the efficiency and specificity of the ITS-PCR amplification as well as the taxonomic resolution of the data. The primers specifically amplified the ITS1 region of *Pythium* s.l. and reduced fungal amplicons. In addition to the ITS region, the cytochrome c oxidase subunit 1 (CO1, COX1) is frequently used in many studies, as it allows the identification of a variety of species of eukaryotic organisms[27,28]. Both ITS and COX1 proved to be highly variable marker regions that distinguish *Pythium* s.l. species[25]. Both regions provide sufficiently high taxonomic resolution for oomycetes, even though both regions fail to detect some species due to barcoding gaps, thus arguing for the inclusion of the ITS region as an efficient barcode for oomycetes identification[25,29].

The cultivation of single species on selective media allows for the generation of individual isolates, which can be used directly for infection experiments and molecular characterization, e.g., by sequencing. However, the selective effects of the medium may influence individual species to a varying degree, causing unequal selection pressure to *Pythium* s.l. as confirmed by ref. 42. A comparison of the soil and bait samples indicates that the selective medium has a certain effect on the genus *Pythium* s.l. Nevertheless, most of the dominant species from the soil samples were also detected as dominant in the bait samples. The selective medium is therefore very useful for detecting dominant species.

### Co-occurrence analysis and linkage of *Pythium* s.l. with soil parameter
Assessing co-occurrence patterns of microbial communities provides a great opportunity to get insights into species-specific interactions, like antagonistic or supportive behavior. However, insights on a global scale are still less understood[43]. Despite no empirical evidence that can

be inferred, co-occurrence networks still support the understanding of species-species interactions, like the ability to colonize specific ecological niches[44]. It is assumed that species that co-occurred often also interact with other taxa more frequently[45] indicating overall importance in microbial communities. In our study, *G. camurandrum* showed the highest number of significant interactions with other species, while negative correlations are dominant (Fig. 7B). Interestingly, *G. attrantheridium* is classified as the most dominant in our study and shows a high degree of negative correlations, implying highly competitive behavior to gain an advantage in growth in the soil. Contrarily, *P. heterothallicum*, the second most abundant species, does show a high number of negative correlations but more significant correlations overall. *G. attrantheridium* is not classified as antagonistic against *Pythium* s.l., raising the question of how manifold the underlying mechanisms are between species.

To link soil properties and the abundance of *Pythium* s.l. The LUCAS topsoil dataset was used to calculate mean values of the included parameters of soil pH and soil organic content. Out of 73 species, only 3 were significantly correlated with soil pH or soil organic content, while 70 species showed no direct correlation, implying an overall balance of the communities across European soils with the ability to react to various soil[46] conditions. This is contrary to previous findings[47] on a large scale, soil pH was the most important bacterial and fungal beta-diversity driver. This discrepancy may reflect the structural stability of *Pythium* s.l. communities in adaptation to various environments.

### High species diversity of *Pythium* s.l. communities in Europe
In this study, we identified a set of ten *Pythium* s.l. species representing the most common and abundant species in Europe. In contrast to a study in Ohio, where *P. dissotocum* and *G. sylvaticum* were the predominant species[3], *P. dissotocum* was not found in Europe, while *G. sylvaticum* was very common. Interestingly, in addition to *P.* aff. *dissotocum* and *G. sylvaticum*, *G. ultimum* var. *ultimum*, *G. heterothallicum*, *G. attrantheridium* and *G. sylvaticum* were found to be predominant in

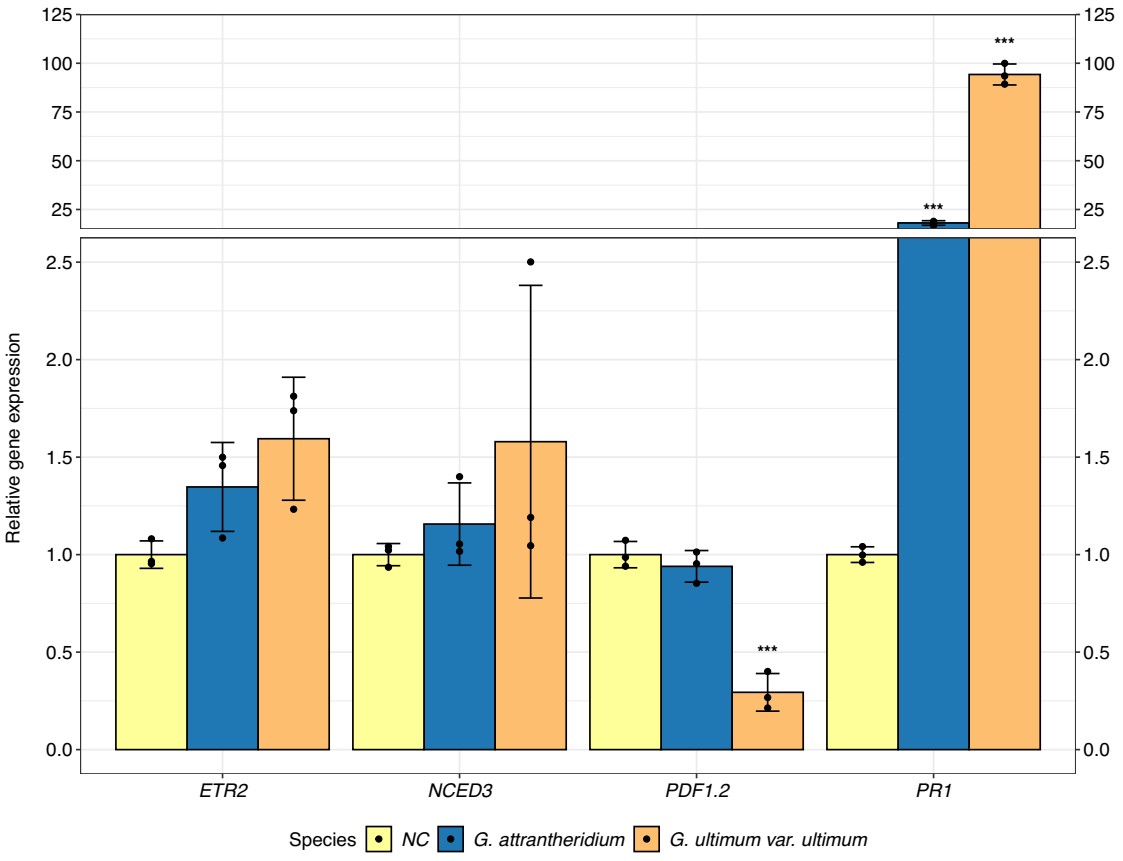

**Fig. 10 | Comparison of gene expression levels of corn in response to *Pythium* s.l. infection.** Effects of *Pythium* s.l. infection on expression levels of main defense-related phytohormone marker genes in corn roots at 7 days past infection quantified by RT-qPCR. *PR1* for salicylic acid (SA), *PDF1.2* for jasmonate (JA), *ETR2* for ethylen (ET) and *NCED3* for abscisic acid (ABA). Relative expression (log$_2$) of each marker gene was displayed (non-infected control (NC), *G. attrantheridium* and *G. ultimum* var. *ultimum*). Statistical differences were tested two-sided by a multiple contrast test for \*\*\**p* < 0.001, \*\**p* < 0.01, \**p* < 0.05 (degrees of freedom = 24). Error bars indicate the standard deviation (*n* = 3).

most soil samples from north America[48]. The discrepancy of varying occurrences of predominant species in different studies may be due to different geographic preconditions and agricultural practices.

Investigations on *Pythium* s.l. diversity of infected soybean roots in the US in 2011 and 2012 showed that in 2011 *G. sylvaticum*, *G. heterothallicum*, *G. ultimum* var. *ultimum* and *G. attrantheridium* were among the 12 most frequently detected species, while in 2012 in addition *G. intermedium* and *G. rostratifingens* were also dominant[48]. Noticeably, four species from 2011 and six from 2012 belong to the ten most frequently detected species in our study, thus strongly suggesting that these species dominate *Pythium* s.l. communities on a global scale. However, our study's relative abundance of *G. sylvaticum* was significantly higher in 2020 than in the other two years. As no significant year-to-year difference in the abundance for *G. attrantheridium*, *G. heterothallicum*, *G. apiculatum*, *G. rostratifingens*, *P.* aff. *hydnosporum*, *P. intermedium*, *P. arrhenomanes* and *G. ultimum* var. *ultimum* was observed, *G. sylvaticum* appeared significantly more abundant in 2020. In addition, *P. monospermum was* also detected at many sites in this study, while the relative abundance of *P. monospermum* was higher in 2019 than in the other two years.

Compared to the mean value of all countries, we found only minor differences in the occurrence of the dominant *Pythium* s.l. species among countries. The average temperatures around sowing time (March to May) vary significantly when comparing countries. Nevertheless, a significantly higher relative abundance of *G. ultimum* var. *ultimum* was found in Belgium and statistic trend for the Netherlands. In contrast, significantly more *G. heterothallicum* was detected in

France and Romania, while *P. monospermum* was significantly more common in Italy. However, it remains open why these differences persist between these countries. A possible scenario could be that the soil samples used in this study tend to represent more of a natural, steady state of *Pythium* s.l. communities that have not yet been sharpened by infection pressure. This also explains discrepancies between our data and other studies, mainly focusing on infected roots. It's very likely that the cropping systems of these two countries, which are characterized by a high percentage of vegetables and different crop rotations, for example, could influence the higher *G. ultimum* var. *ultimum* abundance.

Nevertheless, our data show an even distribution of predominant *Pythium* s.l. species in Europe, regardless of their geographic location and agricultural history, implying a high infection potential of *Pythium* s.l. in Europe. Differences were only observed for soil texture classes. Likewise, the variation of inoculum density in heavy soils is lower than medium and light soils. In general, it should be noted that these soils not only differ in their clay content but are also subject to different cultivation systems due to their characteristics. The reduced tillage, as well as closer crop rotations, could affect soil pathogens by reducing the inoculum of soil-borne pathogens through plowing[49,50]. Another factor could be soil temperature, as lighter soils warm up faster in spring than heavier soils, and the pathogenicity of species is temperature-dependent[48,51,52] and thus can also critically affect inoculum density. Interestingly, soil texture class has an impact on species diversity. For medium soils, diversity tends to be higher compared to the mean of all soil texture classes. This is most likely influenced by the

microbial activity of the soil. The presence and activity of certain bacterial and fungal taxa, as well as the activity of specific enzymes or individual compounds, have an impact on pathogens[53].

## Predominant *Pythium* s.l. is represented by a large number of ASVs

A high number of ASVs within a given species supports the species diversity of *Pythium* s.l. communities in Europe. In particular, *P.* aff. *hydnosporum*, *G. attrantheridium*, *G. rostratifingens*, *P.* sp. *nov*, *G. sylvaticum*, and *G. ultimum* var. *ultimum* were found to have many different ASVs in 2020. Most likely, these differences can be explained by overriding influencing factors such as weather, supported by the European weather data[54].

Although our data do not allow us to conclusively identify the main factor for the different ASVs of the various species, however, soil moisture seems to play a critical role in this process as a prerequisite for zoospore movement in the soil, as initially described by ref. 55. But, based on the biology of *Pythium* s.l. the presence of a host plant is required, as soil moisture is mainly relevant to the initial infection. Thus, a future long-term survey monitoring of the same locations and sampling time points may shed more light on the driving force of *Pythium* s.l. community structure across Europe. Strikingly, certain ASVs were detected more frequently within a predominant species than others, implying their relevance in terms of disease potential. We characterized them by comparing the absolute abundance found at specific locations with the occurrence at all locations. The ASVs most frequently detected within the corresponding species were also most frequently detected at the different sites. Noticeably, this is the case for *P.* aff. *hydnosporum*, *G. apiculatum*, *G. attrantheridium*, *G. intermedium*, *G. rostratifingens*, *G. sylvaticum* and *G. ultimum* var. *ultimum*. All of them are primarily represented by a single characteristic ASV. Partially, this is also found for *G. heterothallicum*, where the number of detections was higher at different sites for ASV_10, although ASV_6 was most abundant in absolute terms for *G. heterothallicum*. It is noteworthy that individual ASVs of certain species occur at a particularly high number of sites, while for other species, distribution is characterized by multiple ASVs. This suggests varying genetic diversities among the different species. The importance of an ASV depends not only on its frequency at different locations but also on its adaptability to a changing environment and agricultural systems. It is generally accepted that the dominance of single genotypes, represented in our study by ASVs, has a selective advantage within a genus, for example, by increasing their pathogenicity and virulence on their hosts.

## *Pythium* s.l. harbors a high disease potential in Europe

We tested *G. attrantheridium* and *G. ultimum* var. *ultimum*, isolated from this study, in an in vitro-infection experiment for their disease potential in corn. Consistent with previous reports[42,56], both species were pathogenic and capable of infecting corn but differed significantly in symptoms and disease progression. The infection of *G. ultimum* var. *ultimum* resulted in much more pronounced disease symptoms and severity on corn roots, causing a significant reduction in germination rate, shoot fresh weight, and root dry weight. Interestingly, although *G. attrantheridium* was the most frequently detected species in this study, the infection with *G. attrantheridium* induced considerably fewer disease symptoms on corn roots and, consequently, a lower-level reduction of shoot fresh weight and root dry weight as well as germination rate than infection with *G. ultimum* var. *ultimum*. These results underline the complexity and importance of corn-Pythium interactions on the disease potential of individual species. Due to climate change, it can be assumed that the distribution and relative abundance of soil-borne pathogens such as *Pythium* s.l. will increase, with implications for food security worldwide[57]. Thus, the distribution, abundance, and pathogenic potential of individual

*Pythium* s.l. should be considered as the main factors determining Pythium-disease potential.

So far, little is known about molecular events in plant response to *Pythium* s.l. Our analyzes highlight transcriptional changes of plant hormone marker genes in response to *Pythium* s.l. infection. Both *G. ultimum* var. *ultimum* and *G. attrantheridium* infections extremely upregulated the expression of *PR1* (SA) while down-regulated *PDF1.2* (JA) suggesting enhanced SA- but suppressed JA-signaling pathways. Generally, SA activates resistance against biotrophic pathogens, while JA is critical for activating defenses against necrotrophic pathogens[35]. SA- and JA-mediated signaling functions are antagonistic and often exploited by pathogens to promote their virulence[36]. As shown in tomato, *Botrytis cinerea* activated SA signaling to suppress JA signaling escaping from the JA-mediated defense responses[58]. JA deficiency in corn and Arabidopsis led to an extreme susceptibility to *Pythium* s.l.[37,59–61]. It is worth mentioning that the change in transcript levels of *PR1* and *PDF1.2* in response to both *Pythium* s.l. infection appeared to correlate with the disease severity observed in this study. Thus, it seems reasonable to conclude that species like *G. ultimum* var. *ultimum*, have evolved an effective virulence strategy to overcome JA-mediated plant defense response by interfering with plant hormone-signaling networks. Further studies are needed to identify the relevant effectors and underlying molecular events that can contribute to a better understanding of the interactions between corn and *Pythium* s.l. and develop new effective control strategies against this pathogen.

## Methods

### Geographic distribution of sampling locations and sampling procedures

For three consecutive years (2019–2021), soil and bait samples were collected from conventionally farmed agricultural fields. A total of 127 sites were sampled in France, Italy, Germany, Belgium, the Netherlands, Spain, Romania, Hungary, the Czech Republic, Switzerland and Austria. Soil samples were collected four weeks after corn seeding. For each site, eight 50 ml soil samples were taken by pressing the 50 mL falcons (Sarstedt, Nürnbrecht, Germany) into the soil with the opening facing downwards. Soil samples were stored at −20 °C. For bait sampling and species cultivation, Benomyl-Ampicillin-Rifampicin-Pentachlornitrobenzol was (Table S1, Supplementary Information) filled into 15 mL centrifuge tubes (Sarstedt, Nürnbrecht, Germany). After solidification of the medium, 1 mm holes were punctured into the tubes using a hot needle. Eight baits were placed in the soil of a sampling site on the day of seeding. Before inserting the tube into the soil, a hole was drilled with an empty tube with 15 cm apart from the corn row. After four weeks, the baits were collected and stored at 4 °C until further processing. The samples were sent refrigerated by express shipping. All used chemicals, materials and kits in this study are provided separately in a file in the appendix (see Supplementary Data 1). An overview of sampling sites is provided in Table S2.

### Isolation of genomic DNA (gDNA) from soil samples

For gDNA isolation from soil, a pooled 5 g sample was used, by mixing the 8 individual samples of one site. For bait samples, 10 ml of eight individual samples of the selective medium were mixed per site. DNA Isolation was performed using a modified CTAB as described by ref. 62 (Table S3). The isolated gDNA was purified using the "NucleoSpin Gel and PCR Clean-up Kit" (Macherey-Nagel, Düren, Germany) according to the manufacturer's instructions. The integrity and quality of the purified DNA were determined by agarose gel electrophoresis, and the DNA concentration was measured semi-quantitatively using lambda DNA of known concentration (10 ng/μL) with ImageLab software (version 4, Bio-Rad Laboratories). Finally, gDNA was stored at −20 °C until further processing.

## DNA library preparation for NGS sequencing

DNA library preparation and NGS sequencing were conducted by Planton GmbH (Kiel, Germany). Briefly, the diverse ITS region among oomycetes[25,63,64] was used. For this study, the ITS1 region was amplified using the primer pair pyth_f (5′–3′; TGCGGAAGGATCATTACCACAC) and pyth_r2 (5′–3′; GCGTTCAAAATTTCGATGACTC), which were specifically designed by generating a multiple sequence alignment of available ITS sequences (NCBI database). Fungal sequences were included to focus on regions offering high dissimilarity. Further, primer specificity and efficiency were finetuned by qPCR with *Pythium* s.l. isolates. PCR was performed using "High Fidelity Platinum SuperFi II PCR Master Mix" (Thermo Fisher, Massachusetts, USA). The DNA quantity and quality were measured by photometric measurement using the "Thermo Scientific Multiskan GO" (Thermo Fisher, Massachusetts, USA) and determined by real-time (SYBR Green) monitored PCR. PCR products were purified using the "NucleoSpin Gel and PCR Clean-up Kit" (Macherey-Nagel, Düren, Germany). DNA libraries were generated using the "NEB Next Ultra II Q5 Kit" (New England Biolabs, Ipswich, USA) and subsequently sequenced on an Illumina MiSeq (Illumina, San Diego, USA) platform, resulting in 250 bp paired-end reads.

## Bioinformatic analysis

The resulting paired-end reads were demultiplexed based on their barcodes and primer sequences were trimmed using Cutadapt[40]. The length of the filtered reads was truncated to 200 bp. Sequencing quality was checked using FastQC[41] and further visually inspected using MultiQC[65]. The remaining reads were analyzed according to the DADA2[66] pipeline including the steps: quality filtering and trimming, learning error rates, identifying sample inference, merging paired-end reads, constructing sequence table, and removing of chimeras. Taxonomic annotation of identified amplicon sequence variants (ASVs) was performed running Blast locally in megablast mode[67]. ASVs were aligned to a curated oomycete ITS database[25] using the best bit score and an identity of higher than 95%. Sequences assigned to *Pythium* sp. *nov.* are a new unknown species in the biological taxonomy. Accordingly, all unknown species were merged under *P.* sp. *nov.* Further, the taxonomically assigned species were annotated to the current genus and clade[68]. The R package "phyloseq" v1.47.0[69] and R (4.4.0)[46] were used for further analysis.

## Linkage analysis between soil properties and the co-occurrence of *Pythium* s.l

The linkage between soil properties and the co-occurrence of *Pythium* spp. s.l. was analyzed using the harmonized open-access topsoil dataset, LUCAS[70]. The European Commission made the dataset available through the European Soil Data Centre and managed by the Joint Research Centre (JRC, http://esdac.jrc.ec.europa.eu/). Based on the locations of our samples, we selected the 5 closest sites from the LUCAS dataset and calculated the mean values of respective soil properties (pH, OC). After merging *Pythium* s.l. species diversity and soil properties, Pearson correlations (two-sided) for each species were calculated based on relative abundance. To determine the co-occurrence patterns of all species in soil, the software FastSpar[71,72] was applied using a generated count matrix and taxonomy annotation as input. First, correlation and covariance measures were calculated using 100 iterations, followed by the calculation of exact *p*-values using 10,000 bootstraps.

## Isolation, cultivation, and characterization of *Pythium* s.l

For isolation of single *Pythium* s.l. species, a piece of selective BARP medium from a single bait was transferred to a Petri dish containing BARP medium. A single hypha was excised under a binocular microscope and transferred onto a fresh BARP medium. This process of hypha excision was repeated five times to ensure isolation. Genomic

DNA isolation was carried out using the CTAB method (Table S4). The ITS region was amplified by PCR using the primer pyth_f and pyth_r2. The PCR Amplicon was cloned into the pGEM®-T vector (Promega, Fitchburg, United States) for Sanger sequencing[73] (Eurofins Genomics GmbH, Ebersberg, Germany).

## Infection experiments on corn

*Pythium* s.l. isolates were cultured on BARP medium at 25 °C for 5 days and stored at 4 °C. Before the infection experiments, freshly grown mycelial plugs of each isolate were transferred to PDA medium and cultivated for 5 days at 25 °C. For each inoculum, 250 g of millet (Bio Gold Hirse, Rewe, Germany) was autoclaved in an Erlenmeyer flask, and then 125 ml of sterile deionized water and thirty plugs (5 mm diameter) of *Pythium* s.l. were added, in which plugs of pure PDA medium served as the non-inoculated control. The Pythium-millet suspension was incubated in the dark at 25 °C for 5 days. The soil for the infection experiments was a mixture of unit soil (Einheitserdewerk ED73, Uetersen, Germany) and sand (Kinderspielsand, Bauhaus, Germany) in a weight ratio of 60 to 40. The Pythium-millet suspension was added to the soil and homogenized. Corn seeds (Benedictio, KWS) were sterilized in 5 % sodium hypochlorite (3 min) followed by 10 % ethanol (1 min). In each pot (9 × 9 cm), 3 corn seeds were planted (3 cm depth) in the inoculated soil. The plants were cultivated for 4 weeks under greenhouse conditions (with a simulated day length of 16 h and temperatures of 20 °C during the day and 18 °C at night. Plants were continuously watered. Shoot fresh and root dry weight (at 65 °C for 2 days) were determined. To visualize the infection of corn roots with species of *Pythium* s.l., sterile glass tubes with open ends (30 cm height × 3 cm diameter) were used, which could be sealed with silicone plugs. Glass tubes were filled with 50 mL of 1/2 MS medium. Sterilized corn seeds were pre-germinated on 1/2 MS medium for 3 days and transferred to the top surface of the MS medium inside glass tubes. Corn seeds were grown for 4 days under short-day conditions (8 h light, 25 °C), and then 2.5 cm diameter mycelial plugs were placed on the bottom side of the MS medium facing towards the roots. The section of the glass tubes containing MS medium was protected from light by covering it with aluminum foil. Seven days post-infection (dpi), the aluminum foil was removed, and photos of the roots were taken with a microscope (ZEISS SteREO Discovery.V20, Carl Zeiss, Jena, Germany).

## Gene expression analysis by RT-qPCR

Corn was cultivated on 1/2 MS medium under short-day conditions (8 h, 25 °C). Mycelial plugs of the isolates *G. ultimum* var. *ultimum* and *G. attrantheridium* were placed adjacent to the roots for infection. Seven days past infection (7 dpi), the roots were harvested. Total RNA from roots of corn was extracted by TRIzol® (Thermo Fisher Scientific, Massachusetts, USA) reagent according to the manufacturer's instructions. A total of 1 μg RNA was treated with RNase-free DNase I (Fermentas, Massachusetts, USA) and further transcribed in a volume of 20 μl with the RevertAid First Strand cDNA Synthesis Kit (Thermo Fisher Scientific, Massachusetts, USA) into first strand cDNA according to supplier's instructions. 2 μl of a 1:10 diluted cDNA preparation was mixed with an 18 μl master mix as described in the manual of the Maxima SYBR Green/ROX qPCR Master Mix (Thermo Scientific, Massachusetts, USA). RT-qPCR was performed on a CFX96 Touch Real-Time PCR Detection System (Bio-Rad, Hercules, USA) using the following conditions: 3 min 95 °C; 45 × 10 sec 95 °C, 10 sec 59 °C, 10 sec 72 °C; 10 sec 95 °C, melting curve from 65 °C to 95 °C. Primers used are listed in Table S5. The gene expression level was determined using the delta $C_T$ Method[74] and calculated in relation to the reference gene *actin* that is stably expressed independent of *Pythium* s.l. infection. The resulting $C_T$ values were normalized by primer efficiency. Each measurement consists of three independent biological replicates incorporating three technical replicates each. The relative expression was

log$_2$ transformed, and the induction was calculated as the difference between *Pythium* s.l. and mock-treated samples.

## Statistical analysis

The statistical software R (4.4.0)[46] was used to evaluate the survey data, and infection- and gene expression trials. The data evaluation started with the definition of an appropriate statistical model[75] with the factors "Year," "Country," and "Soil texture class." A model comparison (main effects model vs. pseudo factor model) proved the interaction effect. The residuals were assumed to be approximately normally distributed and heteroscedastic. For the measurement variable germination rate, the data evaluation started with the definition of a Bayesian generalized linear model[76]. A usual linear model was used for the measurement variables shoot fresh- and root-dry weight per plant. These models included the factor "Species". The residuals followed a binomial distribution for germination rate and a normal distribution for shoot fresh- and root dry weight per plant. The normal distributions of the data evaluations were checked by a graphical residual analysis. For the log-values of the relative gene expression, a statistical model based on generalized least squares[77] was used. The model included the factors "Species" and "Gene" and the interaction term. The residuals were assumed to be approximately normally distributed and homoscedastic. Based on all these models, an analysis of variances (ANOVA) was conducted, followed by multiple contrast tests[78,79] as well as a pseudo $R^2$ was calculated[80] for gene expression (see Supplementary Data 2).

## Reporting summary

Further information on research design is available in the Nature Portfolio Reporting Summary linked to this article.

## Data availability

The raw sequencing data generated in this study have been deposited in the National Center for Biotechnology Information (NCBI) Sequence Read Archive (SRA) under accession code PRJNA1008090. Data belonging to the infection experiment (germination rate, shoot fresh weight, root dry weight and gene expression levels) are available in GitHub: https://github.com/Wilken-Christian-Boie/An-assessment-of-the-species-diversity-and-disease-potential-of-Pythium-communities-in-Europe. The LUCAS topsoil 2018 dataset is freely available and can be downloaded after prior registration at the General-Joint Research Centre (JRC) of the European Commission: https://esdac.jrc.ec.europa.eu/content/lucas-2018-topsoil-data.

## Code availability

All code and source data files are available in GitHub: https://github.com/Wilken-Christian-Boie/An-assessment-of-the-species-diversity-and-disease-potential-of-Pythium-communities-in-Europe.

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

## Acknowledgements

We thank Bettina Bastian for preparing the sampling material and Sören Schneider for DNA isolation. We thank Syngenta staff and farmers for soil sampling and the Institute of Phytopathology at Kiel University for providing PhD scholarship to Wilken Boie. This project was financially supported by Syngenta Crop Protection AG, the Bundesministerium für Bildung und Forschung, Germany (Grant no. 031B0910-A), and the Fachagentur für Nachwachsende Rohstoffe. Germany (Grant no. 221NR-058B).

## Author contributions

W.B. Performed experiments, data generation, and analysis and wrote the first draft of the paper. M.S. conducted data processing and bioinformatic analysis. W.Y. Performed infection experiments and molecular analysis. M.H. supported for statistical analysis. M.G. developed and supervised the project. JV: developed and supervised the project, and D.C.: designed and supervised experiments and finalized the manuscript. All authors reviewed the manuscript draft. All authors read and approved the final manuscript.

## Funding

## Competing interests

The authors declare no competing interests.
