## [Peer Review File · Nature Communications]

An assessment of the species diversity and disease potential of *Pythium* communities in EuropeREVIEWER COMMENTS

Reviewer #1 (Remarks to the Author):

Boie et al provide an assessment of the *Pythium* communities in European crop fields. The scope of this project is the highlight of the study as they sampled from 127 sites across 11 countries. They were able to use both culture independent and dependent methods to assess *Pythium* diversity in the soil. There is however one potentially important issue with their analysis of the meta-barcoding data. On line 212 they state "ASVs which were annotated to the close relative *Globisporangium* were removed from the dataset" This is a significant error, but also a bit confusing. The genus *Globisporangium* is recognized and accepted genus by the Oomycete taxonomy community and includes many of the most important pathogenic species in your study including *P. ultimum*, *P. attractantheridium*, *P. sylvaticum*, *P. heterothallicum* and *P. intermedium*. So discarding *Globisporangium* ASVs may represent the loss of very important data. I suggest the authors re-evaluate the data and try to incorporate this name change into your data. I would also like to point out the *G. ultimum* is now known to comprise multiple species <https://doi.org/10.1080/00275514.2023.2241980>, which begs the questions as to whether the ASV diversity includes these closely related species. Once this issue is resolved I am willing to review the manuscript again to see if the data and results have changed.

The other methodological issue that could be addressed is the effect of temperature on the *Pythium* community. The author indicate that communities differ by country, but in many cases these borders are artificial lines and region in one country may be more similar to a neighboring country than a distant region in the same country. I think including mean annual temperature, or mean temperature at planting may provide a more meaningful assessment of the drivers of *Pythium* diversity. A soil map may also be more appropriate than a political map, given the effect of soil type on structuring of *Pythium* communities.

Overall, this is a very important study but has some issues that need to be addressed prior to being accepted or published.

Below are additional notes and I have added comments to the pdf file.

L 212 – Why were sequences close to *Globisporangium* removed from the dataset?

Globisporangium is part of *Pythium sensu lato*, many of which are pathogenic on corn and soybean. Many of the most abundant species in your study are part of *Globisporangium*.

Figure 6 – Why is the tree rooted on *Pythium sylvaticum*? Why not use an outgroup outside the genus such as *Phytophthora* or *Phytophythium*?

L 390 – Did the phylogenetic analysis use the ASV sequence. This is a truncated portion of a single gene. I would not overinterpret the results of phylogenetic relationships.

L416-418 – This seems overly dramatic as it has been established by many other studies the importance of *P. ultimum* as a pathogen of corn seed and seedlings.

Discussion – When mentioning a publication specifically be sure to write out the references. See text for my notes. There are many of these errors in the discussion.

L457-466 – This discussion is not relevant to this paper as you did not do any experiment on soil sample size for DNA. This is speculation between a study done in Europe vs one done in the U.S. I would remove the entire section.

L467 – I do not understand this sentence. Please edit

L470 – What was the breakdown of % of reads assigned to Pythium, assigned to other Oomycetes, assigned to fungi. It is difficult to judge the validity of these statement without the data.

L489 – Try to avoid just listing species present in one study vs your study. Focus on discussing the relevance of your results. What does it mean? Why does it matter?

Reviewer #2 (Remarks to the Author):

Summary:

This study characterizes the diversity of Pythium spp. across agricultural soils in Europe and their interactions with corn. The authors identified predominant Pythium spp. in soil samples from 127 corn fields sampled across three years using meta-barcoding. They also isolated two common species and tested their capacity to cause disease on corn and influence expression of host genes associated with production of salicylic acid (SA), jasmonic acid (JA), ethylene (ET), and abscisic acid (ABA). Meta-barcoding data showed that certain Pythium spp. were more common in specific European countries and there was variation in intraspecific diversity among the species. Infection of corn plants with *P. atrantheridium* and *P. ultimum* showed only *P. ultimum* suppressed plant growth. Both species increased host expression of a gene linked to SA production, while only *P. ultimum* suppressed a gene associated with JA production. The authors conclude that this study provides essential information for developing strategies for managing diseases caused by these common plant pathogens.

Positive aspects:

This is a comprehensive survey of Pythium species in agricultural soil in Europe. The study also provides novel insight into hormonal changes in plants in response to pythium infection, which is a very poorly studied and understood aspect of these pathogens. The authors provided enough detail in the materials and methods, or additional supplementary data, so that the methods could be reproduced. Appropriate statistical analyses were generally used to analyze data and results were clearly presented in the text and figures. The outcomes of this research will be of interest to individuals working on pathogen ecology and the biology of Pythium and other oomycete pathogens.

Room for improvement:

There are a aspects of the manuscript that could be improved through revision.

- The authors have already done a substantial amount of work sampling and analyzing these data. However, as currently written it seems like they have overlooked opportunities to understand the ecology of *Pythium* communities at the continental scale and highlight the potential novelty of the results. Multiple prior studies (some already referenced) have already shown *Pythium* spp. are very common in agricultural soil, their distribution varies among locations, and species display variation in intraspecific diversity. What makes this study unique? Based on reported data and spatially explicit sampling, there are missed opportunities to determine what underlying ecological processes are driving community assembly (e.g., stochastic processes like dispersal limitation versus niche-based processes related to selection against certain traits). The authors have started to look at this in analyzing soil texture classes. However, there are more appropriate statistical and theoretical frameworks that are commonly used in community ecology (reviewed in: <https://journals.asm.org/doi/10.1128/membr.00002-17>).
- Since relatively few NGS based microbial ecology studies have focused on oomycetes like *Pythium*, there may also be missed opportunities to highlight the technical novelty of the study. However, the authors have not included basic summary statistics such as overall taxonomic distribution of the NGS data and how this compares with the few prior studies using similar approaches to characterize *pythium* communities in soil.
- There is a lot of focus on technical aspects of DNA extractions and amplicon sequencing in the Discussion (e.g., second and third paragraphs). While there isn't anything wrong with this, it seems a little out of place as this study did not address these different aspects of using amplicon sequencing.
- There are multiple spelling and grammatical mistakes throughout the manuscript. Some examples are as follows:
 - o Incorrect use of the formal genus name "*Pythium*" to refer to the disease or a specific pathogen (see inline comments below).
 - o Mixed use of past and present tenses.
 - o See additional Inline comments below.

Inline comments

Title: The title reads a little awkwardly. Should there be "of" between "survey" and "Europe"?

L34: Please change references to "*Pythium* spp." in this sentence to plural terms, rather than singular (e.g., "but their severity are" rather than "but its severity is").

L37: "spp." should not be italicized here or elsewhere.

L38: Please change "We identified also" to "We also identified" to improve grammar.

L40: Please clarify what "a dynamic sharpening process" means.

L52: "*Pythium*" is a genus that includes species that are necrotrophic soil-borne plant pathogens.

L57: Similar to prior comment, "*Pythium*" is a genus, not a disease. It might be more appropriate to

use the term “pythium” (unitalicized), to refer to the disease.

L75-88: The use of meta-barcoding is fairly common knowledge. Are there any meta-barcoding focused on oomycetes that you could introduce in this paragraph. Alternatively, citing a lack of this work on oomycetes would emphasize the novelty of your research.

L122-123: Please provide an estimate of the volume or mass of soil in each sample, if known.

L180: Please insert a comma between “was” and “used” to improve grammar.

L182-183: Please include a citation for these Pythium-specific primers. If these are novel primers, please include information on how they were selected.

L218: Should read “Corn plants were” or “Corn was”.

L219: Please clarify why these species were used for the experiments. Was this because they were most common?

L231: Please add a citations for the primers listed in Table S2.

L252-253: Please include which post-hoc test was used to make these contrasts.

L256-263: It would be helpful to include some summary statistics about sequence output and classification here or as supplementary information. For example, how many total sequences, average depth per-sample, and percent of data classified as Pythium.

L267-269: Please clarify what data in parentheses represents. Are these the number of samples a given species was detected in?

L280: Please clarify what how “P. sp.” were defined, either here or in the Materials and Methods. Are these multiple different unidentified Pythium spp. grouped together or are all sequence data aligning to a single unidentified Pythium sp.?

L334: Please clarify use of the terms “compositional” and “structural” diversity. Are these redundant?

L340: Please clarify how inoculum density was measured. Is this from culturing on BARP?

L448: Is there a general summary statement you could include at the end of this paragraph that links your study with these prior studies?

454-465: This content focused of technical aspects of using different amounts of soil seems a little out of place since this was not a goal or something that was addressed in this study.

L463: Please clarify what “more scattering results” means.

L467: Please consider revising this sentence to improve clarity. “Taxonomic resolution is relying” reads awkwardly. A more appropriate phrase would be “Taxonomic resolution depends on...” or some equivalent.

L468-469: Please clarify what was improved in this study. What does “efficiency” refer to in this context? It would also be helpful to include some results showing the specificity of the primers used in this study, so they can be compared to prior ITS-amplicon sequencing studies. “Data” are plural.

L481: Please change “being” to “was” to improve grammar.

L508: Please change “between” to “among” if you are comparing more than two countries.

L519: “data” are plural.

L521-522: Higher infection potential in Europe, compared to where? Significant differences in what?

L525: Please clarify what “natures” means in this context. Are you referring to the chemical or physical properties of the different soils?

L527-538: This section is a little unclear. Are you saying soil type (i.e., environmental filtering) or species interactions are more important in structuring *Pythium* communities?

L540: Please clarify how “compositional” and “structural” diversities are different.

L541-543: Please clarify this sentence. It sounds like you are discussing population level processes (i.e., diversity within species), but refer to *Pythium* “communities”.

L544-546: Please clarify this sentence. What are “these differences” referring to. Are you suggesting weather is the primary factor influencing diversity within *Pythium* spp? How do results from this study support this conclusion?

L547: “data” are plural.

551-553: Why is it surprising/”striking” that are more common ASVs within species? Presence of common and rare genotypes and species is a common characteristic of biological populations and communities. Why do you assume a more common ASV is more functionally relevant and what specific function are you referring to?

L561-563: Please clarify this sentence.

L563-565: Please clarify this sentence. Are you referring to dominance of a genotype/ASV within species, not “genus”?

L566-575: As written, this is describing what a systematics study would focus on. Is this content important to the study? If it is included, it may be valuable to try to link the phylogenetic relationships among the common species to their ecology.

L587-589: Please consider revising this sentence to improve clarity. In particular, it is unclear what “is likely varied, with implications for food security worldwide” means. Are you suggesting that climate change will alter the distribution of different *Pythium* spp. in soil?

L589-591: Please clarify how “occurrence” and “distribution” differ in this context.

L589: Should “antagonistically” read “antagonistic”?

L610: Please change “is” to “are” and “its” to “their” since *Pythium* spp. is a plural term.

Reviewer #3 (Remarks to the Author):

This manuscript describes the diversity of *Pythium* species recovered from soil sampled from corn fields across Europe. The authors also evaluated the pathogenicity of two species on corn. Although this appears to be an excellent study that will contribute greatly to our understanding of *Pythium*, my greatest concern is that the authors have overreached with their interpretation of the data, especially in terms of disease potential. Presence of a pathogen does not necessarily mean disease. There are no data provided if seedling disease was present in the fields sampled. Although the authors evaluated the pathogenicity of two species, they only tested at one temperature. Matthiesen et al 2016 and Rojas et al 2017 both showed temperature affects virulence of *Pythium* species. Moreover, it is not clear from the Methods, how many isolates of each species were evaluated.

I have several concerns, suggestions and edits that can be found in the attached pdf.

- More details are required in the Materials and Methods. I would not be able to repeat your study.
- Data are only different if they are statistically different ($P < 0.05$). Throughout the results, there are instances of “increasing tendency”, “varied to some extent”, “tended to be”, “a slight difference”, etc. Science is very exact. Either it increased or it didn’t based on what the statistics detect.
- You collected soil samples, presumably for DNA extraction, and also baited *Pythium* species from the soil using falcon tubes containing selective medium. In the results, it is not clear to me, if the number of *Pythium* species you report are from the soil samples-DNA sequencing or the soil baiting-isolation or both (summed together). Along this line, were differences between the *Pythium* species detected by sequencing and the *Pythium* species recovered by baiting. This would be very interesting information.

Below are some line-by-line comments. Please also refer to the attached pdf.

Abstract

Needs to be rewritten. I think you could include more details

Line 34-35 spp. is plural therefore please edit the first sentence. *Pythium* spp. pose a serious threat to agricultural production worldwide, but their severity is often neglected because little knowledge about them are available.

Line 37 delete “As a result,”

Line 38-40 Please edit sentence for clarification. I think you could include more details, for example, range of the numbers of species recovered in each country. You could also include that 10 species were common across all countries.

Line 41-42 I disagree that the *Pythium* species differ “Strongly”. See my comments below.

Lines 43-45 The gene expression data could be a different sentence. Also why only mention JA defenses when the relative change in expression of PR1 was 20-100 fold higher than the non-inoculated control?

Line 47-48 I disagree that your work contributes to our understanding of the evolution of virulence and pathogenicity. And I have to question if it contributes to plant-pathogen interactions, since the species you recovered were present in the soil, and no data are presented on species that were present in corn plants.

Introduction

See pdf for editorial suggestions

Line 67 Please use virulence rather than pathogenicity. “moderate pathogenicity” – according to APS, pathogenicity is the ability to cause disease. An organism can either cause disease or not. Virulence is a degree or measure of pathogenicity; the relative capacity to cause disease.

Note that Matthiessen et al. 2016 showed *P. sylvaticum* and other species had high virulence on corn.

Line 70-72 This sentence seems out of place. And perhaps should be included in the discussion with reference to the pathogenicity tests you did.

Line 110-112 what about the SA-mediated response?

Materials and Methods

Significant details were missing. I am not sure I would be able to repeat your study. See pdf for editorial suggestions and additional comments.

Line 116-130 Soil samples were collected 4 weeks after corn seeding and placed in a 50 ml tube – how did you collect the soil samples? Shovel/soil probe? How many samples were collected from each site to place in the 50 ml tube? Did you collect the soil samples from the plant row? Did you mix the soil and then place a sample of it in the 50 ml tube? What was this soil used for? I presume DNA extraction but it is not clear from your methods.

Line 121-123 what do you mean by “dominantly characterized by intensive cultivation of corn”? is corn grown in those fields every year? Is it rotated with something? Table S12 (Should be Table S1, since it is the first supplementary table presented) does not have precrop (do you mean previous crop) or tillage data. I agree these may be helpful data to include so if you do have those data I'd like

to see them in the revision.

Line 123-129 When were the baits placed in the soil? So, they were 15cm from the plant row. This is quite a distance from the corn seedlings, so they were likely sampling *Pythium* species present in the bulk soil, rather than the rhizosphere soil. How were the baits spaced apart from each other? How many holes were in each bait tube? Where were the holes positioned on the tube – at different heights? Different sides?

Lines 141-169 Of all the species, why did you choose to evaluate the pathogenicity of *P. attractantheridium* and *P. ultimum*? How many isolates of each species did you evaluate? Please use ‘inoculated’ rather than ‘infected’ – you inoculated corn with the pathogen, and then evaluated infection. Inoculation does not always cause infection. What was the ratio of *Pythium*-colonized millet:soil substrate? I do not understand this sentence “Because of minimal soil residues, an exact measurement of root fresh weight was not possible.”

Line 159-168 I am having a hard time visualizing what you did here. Why were the corn seeds grown under short day conditions?

Line 170-171 Unclear. Were the 8 individual soil samples all in 8 individual 50 ml tubes, that were mixed and then a 5 g subsample removed? If yes, this information should be in the soil sampling section. In this line suggest you rewrite the sentence “For gDNA isolation, a 5g soil sample was removed from the 50ml Falcon tube sample of soil collected from each sampled field”.

Line 180-181 Not sure what you mean “was used justified by and a”. How did you “optimize” the primer pairs? Did you design the primer pairs? If not, please provide a reference.

Line 186 “quantity and quality “of what?

Line 207 If there are 171 *Pythium* species they must all be different. “different” is redundant – please delete here and throughout the manuscript.

Line 218-222 More details are required on how you prepared the plants, and inoculated them for RT-q-PCR analysis. Please see pdf.

Line 231 Please include the name of the authors and then the superscript for the reference, e.g. Pfaffel⁵². There are several cases of this in the discussion too.

Results

In the results, usually only observations are reported. Throughout this section there are sentences that pertain to the Materials and Methods, or Discussion. Please check the attached pdf.

Line 283-284 This sentence should be in the Discussion

Line 285 & 287 What is the difference between “sampled sites” and “sampled locations”? Be consistent with terminology.

Line 284 Delete “Additionally,”

Line 292-300 Lots of redundancy, I have made some suggestions to the text.

Line 303-312 I disagree with your statements. There are no statistical differences among years for these species, and therefore there was no difference in relative abundance for these species among years. Please rewrite.

Figure 2 Please insert years, move the taxa legend to the far left since that is the first heat map we see.

Figure 3 No significant differences (*) are indicated. I inserted them from Table S3. Please check. What method did you use to detect statistical differences? LSD? Tukeys? Please include this information in the Methods text as well.

Line 325 No idea what you mean by “latent”

Line 340-350 Since soil texture did not affect *Pythium* inoculum density, these sentences need to be deleted. I have suggested a sentence that can be used on the attached pdf.

Line 350-352 please clarify.

Line 359 “higher diversity” – where is the statistics to prove this?

Line 371 “difference is ASV counts” – statistics?

Figure 5 I wonder if the data for All years could be shown as a bar graph. It might be easier to compare among species?

Line 376, 379-380 please clarify.

Figure 6B and line 390-402 It is not clear to me what the purpose of the phylogenetic analysis was in your study. Did you observe different phylogenetic relationships to what has been reported before among *Pythium* species?

Line 410-411 did you not assess root rot severity? Fig. 7C, D and E are okay but a bar chart of root rot with SE bars and statistics support your statement “more severe disease symptoms”. Also these are stereomicroscopic images, right?

Line 410-415 I like to include the exact P-value (to 2-3 decimal places) when I am stating differences. I am not sure if Nature Communications likes that so it would be worth checking some recent publications to see if exact P-values are used. If yes, then please use them throughout the results.

Figure 8 Spell out everything in the Figure legend. *** = $P < 0.001$? – please include in legend.

Discussion

I like this link to how to write a discussion – particularly the first paragraph

<https://www.biosciencewriters.com/How-to-Write-a-Strong-Discussion-in-Scientific-Manuscripts.aspx>

Line 440-442 belongs in results “*Pythium* was detected in the soil at 125 of 127 locations.

Line 438-477 need to be reworked. The discussion goes back and forth between the DNA sequencing and isolation on selective media and consequently is difficult to follow. If you include data on the *Pythium* species detected by sequencing, and the *Pythium* species recovered by baiting, this would help with your discussion and the points you make.

Line 463 not sure what you mean by “scattering”

Line 469-470 how did you optimize the primers? Why did they reduce fungal amplicons?

Line 478 This is not genetic diversity – the paragraphs below discuss species diversity.

Line 489 Broders et al 2007, Jaing et al 2012, Matthiesen et al 2016, Rojas et al. 2019 reported *Pythium* spp. diversity in corn roots so I am curious why you chose to compare your data to Rojas et al. 2017 which describes *Pythium* spp. recovered from soybean? I would encourage you to do a more thorough discussion of *Pythium* spp. that have been reported from corn. Bickel and Koehler 2021 have a good review.

Line 512 what do you mean by ‘particularly pronounced’.

Line 515-516 – This needs to be expanded on. Moreover, this is a shortcoming of your study: you have reported *Pythium* species present in the bulk soil of corn fields; it is possible there is less diversity of *Pythium* species in the rhizosphere, and within symptomatic corn rots.

Line 520 You did not report data for ‘agricultural history”

Line 521 Significant differences among soil texture classes were only detected for number of

species detected (or recovered).

Line 522 Delete. Inoculum density was not significant for soil texture.

Line 533-538 There are no data to support soil suppressiveness. Delete.

Line 543-546 This needs further explanation. What are the rationales for proposing this?

Line 566-575 I am not familiar with Pythium phylogeny and I suspect most of your readers will not be too. Is this any different than the relationships that have been described before? Again, I don't think these data add anything to your study. I would far rather see a comparison of the Pythium species detected in the soil by DNA sequencing or recovered by baiting.

Line 582-586 I think this statement is overreaching. Who else has reported *P. attractantheridium* as a pathogen of corn? Is it possible that this species could cause more severe disease at lower temperatures compared to what you tested it at (20°C)? Rojas et al and Mathhiesen et al both report species that are high virulent at 13°C and significantly less so at 18°C and/or 23°C.

Line 588-589 Please clarify. What are the rationales for suggesting this?

Line 592-608 I'd like to see more of a discussion regarding the strong expression of PR1.

Line 615 "with great disease potential" - based on what data?

Supplementary materials

Table S1. Was this protocol completely developed by you or did you modify an existing protocol? If the latter, the existing protocol should be referenced. Check spelling of ammoniumbromid – think it ends in 'e'

Table S2. Delete "Primers used in this study." Redundant with next sentence. Do these need to be referenced?

Table S3 (and all other pertinent tables). Please indicate what o, *, ** and *** are? Presume P<0.1, 0.05, 0.01 and 0.001 respectively? And what statistical test you used.

Table S4. Netherlands - insert 's'

Table S5. Edit last phrase of heading: (* designation of the soil type according to ONORM L 105080). Insert e in Soiltyp* heading of the table.

Table S6. Modify title "Abundance of Pythium estimates based on soil texture classes (heavy, medium and light) compared to the mean of Pythium abundance for all sites sampled."

Table S7. Modify title “Estimates of different Pythium species based on soil texture classes (heavy, medium and light) compared to the mean number of Pythium species for all sites sampled.”

Table S8. Modify title “Comparison of the germination rate of corn inoculated with Pythium attrantherdium and P. ultimum with a non-inoculated control. * indicates significance at $P < 0.05$ based on what statistical test?

Table S9. Modify title “Comparison of the shoot fresh weight per plant of corn inoculated with Pythium attrantherdium and P. ultimum with a non-inoculated control. * indicates significance at $P < 0.05$ based on what statistical test?

Table S10. Modify title “Comparison of the root dry weight per plant of corn inoculated with Pythium attrantherdium and P. ultimum with a non-inoculated control. * indicates significance at $P < 0.05$ based on what statistical test?

Table S11. Modify title “Comparison of gene expression of PR1, PDF1.2, JA, ETR2 and NCED3 in corn seedlings inoculated with Pythium attrantherdium (P2), P. ultimum (P38) or non-inoculated (NC). *** indicates differences between expression levels at $P < 0.001$ using a two sided xxx test.

Table S13. Was this protocol completely developed by you or did you modify an existing protocol? If the latter, the existing protocol should be referenced.

Response to Reviewers to Manuscript ID: NCOMMS-23-42861

To Reviewer #1 (Remarks to the Author)

Main remark:

We have re-analyzed and re-evaluated our database together with the genus *Globisporangium* as suggested. Since *Globisporangium* spp. are not among the 10 most abundant species across all years, they do not affect our main results. A point-by-point response/amendment can be found in the revised manuscript, which is highlighted in yellow.

Question: L 212 – Why were sequences close to Globisporangium removed from the dataset? Globisporangium is part of Pythium sensu lato, many of which are pathogenic on corn and soybean. Many of the most abundant species in your study are part of Globisporangium.

Answer:

For this survey, no any *Globisporangium* sequence has been removed from the database. We generated a database based on the ITS1 sequences of the genus *Pythium* available on NCBI by utilizing the research tool to request all sequences associated with the term “*Pythium* internal transcribed spacer”. During this, sequences referred to as *Globisporangium* were also downloaded, most probably based on the close relationship of both taxa. However, a taxonomic assignment for *Globisporangium* do not reflect our primary project focus. We know that individual ASVs can be assigned to *Pythium* and *Globisporangium* due to their sequence similarity in the ITS region. Following the reviewer’s suggestion, we included the *Globisporangium* spp. from the dataset and re-analyzed our data. Since *Globisporangium* spp. are not among the 10 most common species across all years, they do not show any impact on our primary results. We have clarified this and added corresponding information in M&M, lines 166-171:

“The final database consists of 11.431 sequences containing 8.342 *Pythium* and 3.075 *Globisporangium* sequences. Resulting annotations were filtered, matching a minimum percent identity of 95 % and a minimum e-value of 1E-5. Although since 2010, the genus *Globisporangium* was segregated⁴⁶, we decided to keep *Globisporangium* records in the database as both share a high sequence similarity in the ITS region.” and in the results section, lines 263-271: “In total, 82 different *Pythium* spp. were detected, from which 13, however, can be annotated as *Globisporangium* spp.”.

Question: Figure 6 – Why is the tree rooted on Pythium sylvaticum? Why not use an outgroup outside the genus such as Phytophthora of Phytophythium?

Answer:

Yes, it could be that. As the tree does not provide a meaningful message to this study, we have decided to remove it from the revised version.

Question: L 390 – Did the phylogenetic analysis use the ASV sequence

Answer:

Yes. Because the tree was not based on the complete genomes, its value is limited. Therefore, we decided to substitute it by co-occurrence analysis and linkage analysis with soil properties. This change may also justify a critical objection of the reviewers.

Question: L416-418 – This seems overly dramatic as it has been established by many other studies the importance of P. ultimum as a pathogen of corn seed and seedlings.

Answer:

Our data show again that *P. ultimum* has a higher disease potential than, for example, *P. atrantheridium* in European corn fields, which is consistent with other studies. We have removed this sentence from the results and integrated it into the discussion.

Question: Discussion – When mentioning a publication specifically be sure to write out the references. See text for my notes. There are many of these errors in the discussion.

Answer:

We fully agree with the reviewer. But this type of citation is defined by NC, we could not change it. However, we corrected corresponding text.

Question: L457-466 – This discussion is not relevant to this paper as you did not do any experiment on soil sample size for DNA. This is speculation between a study done in Europe vs one done in the U.S. I would remove the entire section.

Answer:

We agree. We have removed this part of the discussion as the reviewer suggested.

Question: L467 – I do not understand this sentence. Please edit.

Answer:

Sorry for this ambiguity. To clarify it, we changed the sentence in line 428: "Taxonomic resolution depends on the used primers and the sequenced marker region."

Question: L470 – What was the breakdown of % of reads assigned to Pythium, assigned to other Oomycetes, assigned to fungi. It is difficult to judge the validity of these statement without the data.

Answer:

For this study, we generated a reference database only with Pythium and Globisporangium sequences annotated/available in NCBI, and furthermore, we used specially designed Pythium ITS-primers for PCR reaction, so that assignment of reads to other organisms was minimized. We clarified this point in lines 430-431: "The primers specifically amplified the ITS1 region of Pythium and reduced fungal amplicons." And in lines 140-146: "For this study, the ITS1 region was amplified using the primer pair pyth_f (5'-3'; TGCGGAAGGATCATTACCACAC) and pyth_r2 (5'-3'; GCGTTCAAATTTTCGATGACTC), which were specifically designed by generating a multiple sequence alignment of available ITS sequences (NCBI database). Fungal sequences were included to focus on regions offering high dissimilarity. Further, primer specificity and efficiency were finetuned by qPCR with Pythium isolates."

Question: L489 – Try to avoid just listing species present in one study vs your study. Focus on discussing the relevance of your results. What does it mean? Why does it matter?

Answer:

Many thanks. We have tried to improve this part in lines 484-502.

Reviewer #2 (Remarks to the Author)

Main remark:

As suggested, we have tried to analyze our data from a community ecology perspective. We used the software "FastSpar" to determine the interactions/correlations between species. Overall, negative correlations were predominant for all 82 identified species (2257 negative, 1146 positive). While *P. attrantheridium* showed the highest number of significant correlations (82), *P. graminicola* exhibits the lowest. Further, we included general soil parameters (pH, soil organic content) into the data analysis by using the LUCAS topsoil dataset made available by the European Commission through the European Soil Data Centre and managed by the Joint Research Centre (JRC, <http://esdac.jrc.ec.europa.eu/>). Out of 82 species only 5 species were significantly correlated with soil pH or soil organic content, while 77 species showed no direct correlation, implying an overall balance of the communities across European soils with ability to react to various soil conditions. A point-by-point response/amendment can be found in the revised manuscript, which is highlighted in yellow.

Question: Title: The title reads a little awkwardly. Should there be “of” between “survey” and “Europe”?

Answer:

We have changed title to “Pythium Survey of Europe provides insights into the species diversity of *Pythium* communities and their disease potential in Europe”.

Question: L34: Please change references to “Pythium spp.” in this sentence to plural terms, rather than singular (e.g., “but their severity are” rather than “but its severity is”).

Answer:

we have changed it in line 35: “*Pythium* spp. pose a serious threat to agricultural production worldwide, but their severity is often neglected because little knowledge about them are available.”, and as well as through the manuscript.”.

Question: L37: “spp.” should not be italicized here or elsewhere.

Answer:

We have corrected this sentence as well as throughout the whole text as suggested.

Question: L38: Please change “We identified also” to “We also identified” to improve grammar.

Answer:

Yes, it has been changed as suggested.

Question: L40: Please clarify what “a dynamic sharpening process” means.

Answer:

Because we have only three-year data, it is not possible to find out the driving force for “a dynamic sharpening process”. This study, however, shows a high species diversity of *Pythium* communities across European corn fields, which should be a result of a sharpening process occurred. To understand this process, further studies are needed. To avoid misunderstanding, we decided to remove this sentence from the abstract.

Question: L52: “Pythium” is a genus that includes species that are necrotrophic soil-borne plant pathogens.

Answer:

We changed it in line 52 to “*Pythium*” is a genus that includes species that are necrotrophic soil-born plant pathogens with more than 300 species.”.

Question: L57: Similar to prior comment, “Pythium” is a genus, not a disease. It might be more appropriate to use the term “pythium” (unitalicized), to refer to the disease.

Answer:

Thanks, we have changed it in the sentence as well as in the revised version thoroughly.

Question: L75-88: The use of meta-barcoding is fairly common knowledge. Are there any meta-barcoding focused on oomycetes that you could introduce in this paragraph. Alternatively, citing a lack of this work on oomycetes would emphasize the novelty of your research.

Answer: Thanks for your highlighting this point.

Although there was a report from the USA in which the Cox region was used to detect *Pythium* (Rojas et al. 2019) that we also cited, in our knowledge, our study is the first NGS meta-barcoding of *Pythium* based on the ITS region on such a large scale. As the review suggested, we highlighted this point in abstract lines 46-50: “This study provides a comprehensive and valuable metabarcoding dataset that enables deep insights into the species diversity of *Pythium* communities in European corn fields and the basis for a better understanding of plant-*Pythium* interactions, facilitating the development of an effective strategy to cope with this pathogen in the future. “.

Question: L122-123: Please provide an estimate of the volume or mass of soil in each sample, if known.

Answer:

Eight soil samples were taken from each site, each with a volume of 50 ml. The entire Falcon was filled with soil. We added this information in lines 128-129: "For gDNA isolation from soil, a pooled 5 g sample was used, by mixing the 8 individual samples of each site. For bait samples, 10 ml of eight individual samples of the selective medium were mixed per site."

Question: L180: Please insert a comma between "was" and "used" to improve grammar.

Answer:

We have made the corresponding change in line 139: " Briefly, the diverse ITS region among oomycetes^{23,38,39,39} was used, based on a large number of reference ITS sequences available on NCBI Genbank (December 2022)."

Question: L182-183: Please include a citation for these Pythium-specific primers. If these are novel primers, please include information on how they were selected.

Answer:

The primers used were specially developed for this study. We clarified this in line 140-146: "For this study, the ITS1 region was amplified using the primer pair pyth_f (5'-3'; TGCGGAAGGATCATTACCACAC) and pyth_r2 (5'-3'; GCGTTCAAAATTCGATGACTC), which were specifically designed by generating a multiple sequence alignment of available ITS sequences (NCBI database). Fungal sequences were included to focus on regions offering high dissimilarity. Further, primer specificity and efficiency were finetuned by qPCR with Pythium isolates."

Question: L218: Should read "Corn plants were" or "Corn was".

Answer:

We changed it in line 222: "Corn was cultivated on 1/2 MS medium under short-day conditions (8 h, 25 °C)."

Question: L219: Please clarify why these species were used for the experiments. Was this because they were most common?.

Answer:

Yes, the two species (*P. ultimum* and *P. attrantheridium*) are very common in Europe. And, most importantly, the isolates are confirmed by comparing the Sanger sequencing data of the isolates with their dominant ASVs detected by NGS. We added this information in lines 222-224: "Mycelial plugs of the isolates *P. ultimum* and *P. attrantheridium*, which are widespread in Europe, were placed adjacent to the roots for infection."

Question: L231: Please add a citation for the primers listed in Table S2.

Answer:

All primers used in this study were designed by ourselves. They are listed in Table S5. We added this in the M&M, lines 234-237: "Primers used are listed in Table S5. Gene expression level was determined using the delta C_T Method⁵³ and calculated in relation to the reference gene *actin* that is stably expressed independent on Pythium infection."

Question: L252-253: Please include which post-hoc test was used to make these contrasts.

Answer:

The multiple comparisons were performed using an appropriate user-defined contrast for the corresponding influencing variables with the "multcomp" package in R. According to Hothorn, Bretz, and Westfall (2008). We state this point in line 255-257: "Based on all these models, an analysis of variances (ANOVA) was conducted, followed by multiple contrast tests^{57,58} as well as a pseudo R² was calculated⁵⁹ for gene expression."

Question: L256-263: It would be helpful to include some summary statistics about sequence output and classification here or as supplementary information. For example, how many total sequences, average depth per-sample, and percent of data classified as Pythium.

Answer:

We have indicated in lines 153-154: “Sequencing statistics are provided in the electronic supplementary material.”.

Question: L267-269: Please clarify what data in parentheses represents. Are these the number of samples a given species was detected in?

Answer:

We clarified this in lines 264-271: “Noticeably, a single soil sample (P20_L15) contained up to 21 species while no Pythium was detectable in two samples (P20_L07 and P20_L37, Figure 2). As illustrated by the heatmaps and the annotation of the data from three sampling years (Figure 2), multiple species were profoundly dominant, 52 different Pythium species were detected in 2019, from which the 10 most common species were *P. attrantheridium* (32), *P. heterothallicum* (31), *P. sylvaticum* (31), *P. monospermum* (29), *P. sp* (27), *P. oligandrum* (25), *P. ultimum* (21), *P. rostratiformis* (19), *P. intermedium* (18) and *P. aristosporum* (18).”.

Question: L280: Please clarify what how “P. sp.” were defined, either here or in the Materials and Methods. Are these multiple different unidentified Pythium spp. grouped together or are all sequence data aligning to a single unidentified Pythium sp.?

Answer:

In our knowledge, *P. sp* consists of different undefined *Pythium* species, which were taxonomically assigned according to the reference database based on NCBI. We indicated it in line 270: “*P. sp* (27, undefined *Pythium* species)”.

Question: L334: Please clarify use of the terms “compositional” and “structural” diversity. Are these redundant?

Answer:

We preferred to use the term “species diversity” instead of “compositional” and “structural” diversity, as suggested by reviewers, in the manuscript.

Question: L340: Please clarify how inoculum density was measured. Is this from culturing on BARP?

Answer:

We have clarified this point in lines 327-329: “We defined Pythium density by the total number of reads detected in the soil sample of a respective site. Interestingly, the light soils showed the highest Pythium density, followed by the medium and heavy soils.”.

Question: L448: Is there a general summary statement you could include at the end of this paragraph that links your study with these prior studies?

Answer:

Thanks for the suggestion. We added in in lines 426-427: “Thus, this study offers a comprehensive and valuable barcoding dataset for further studies on *Pythium* spp.”.

Question: 454-465: This content focused of technical aspects of using different amounts of soil seems a little out of place since this was not a goal or something that was addressed in this study.

Answer:

We agree. This part has been removed from the discussion.

Question: L463: Please clarify what “more scattering results” means.

Answer:

According to the authors Morita and Akao (2021) the Shannon-Wiener index is different for DNA isolations from 5 g of soil than from smaller amounts. The Shannon-Wiener Index is a mathematical

quantity that is used to describe diversity. It therefore describes the diversity in the samples, taking into account both the number of different taxa and the abundance.
Due to the argument in the previous point, this has been removed from the discussion.

Question: L467: Please consider revising this sentence to improve clarity. "Taxonomic resolution is relying" reads awkwardly. A more appropriate phrase would be "Taxonomic resolution depends on..." or some equivalent.

Answer:

Yes, we have rewritten the sentence in line 428: "Taxonomic resolution depends on the used primers and the sequenced marker region."

Question: L468-469: Please clarify what was improved in this study. What does "efficiency" refer to in this context? It would also be helpful to include some results showing the specificity of the primers used in this study, so they can be compared to prior ITS-amplicon sequencing studies. "Data" are plural.

Answer:

We clarified in lines 429-431: "Application of Pythium specific primers greatly increased the efficiency and specificity of the ITS-PCR amplification as well as the taxonomic resolution of the data. The primers specifically amplified the ITS1 region of Pythium and reduced fungal amplicons."

Question: L481: Please change "being" to "was" to improve grammar.

Answer:

Changed in lines 477-478: "Differing from a study in Ohio, who found that both of *P. dissotocum* and *P. sylvaticum* were the predominant species⁸."

Question: L508: Please change "between" to "among" if you are comparing more than two countries.

Answer:

We have made the corresponding change in line 504: "In comparison to the mean value of all countries, we found only minor differences in the occurrence of the dominant *Pythium* species among countries."

Question: L519: "data" are plural.

Answer:

It has been changed in line 514: "Nevertheless, our data show an even distribution of predominant *Pythium* species in Europe."

Question: L521-522: Higher infection potential in Europe, compared to where? Significant differences in what?

Answer:

We have rewritten this sentence to clarify that the disease potential of *Pythium* is in general high in Europe in lines 517-520: "Nevertheless, our data show an even distribution of the predominant *Pythium* species in Europe, despite the different geographical location and agricultural practices/cropping systems, suggesting a high infection potential of *Pythium* in Europe. Significant differences were only observed for soil texture classes."

Question: L525: Please clarify what "natures" means in this context. Are you referring to the chemical or physical properties of the different soils?

Answer:

We replaced "natures" with "characteristics" line 522: "In general, it should be noted that these soils not only differ in their clay content but are also subject to different cultivation systems due to their characteristics."

Question: L527-538: This section is a little unclear. Are you saying soil type (i.e., environmental filtering) or species interactions are more important in structuring Pythium communities?

Answer:

Yes. We have rewritten this section in **lines 524-530**: “Another factor could be soil temperature, as lighter soils warm up faster in spring than heavier soils and the pathogenicity of species is temperature dependent^{61,74,75} and thus can also critically affect inoculum density. Interestingly, soil texture class also has an impact on diversity. For medium soils, diversity is significantly higher compared to the mean of all soil texture classes. This is most likely influenced by the microbial activity of the soil. The presence and activity of certain bacterial and fungal taxa as well as the activity of specific enzymes or individual compounds may have an impact on pathogens⁷⁶.”.

Question: L540: Please clarify how “compositional” and “structural” diversities are different.

Answer:

After due discussion, we decided to use “species” diversity instead of “compositional” and “structural” diversities throughout the manuscript in **lines 534-535**: “A high number of ASVs within a given species supports the species diversity of Pythium communities in Europe.”.

Question: L541-543: Please clarify this sentence. It sounds like you are discussing population level processes (i.e., diversity within species), but refer to Pythium “communities”.

Answer:

We tried to explain the observed diversity within a given species. But, we do not have sufficient data to strengthen this. To avoid misunderstanding, we decided to remove this sentence in the revised version.

Question: L544-546: Please clarify this sentence. What are “these differences” referring to. Are you suggesting weather is the primary factor influencing diversity within Pythium spp? How do results from this study support this conclusion?

Answer:

“These differences” refer to “a higher number of different ASVs of within *P. rostratiformis*, *P. sylvaticum* and *P. ultimum* in 2020. However, the limited amount of data does not allow the driving force to be defined conclusively. For this reason, we have decided to delete this sentence in the revised version. We rewritten the sentence in **lines 536-539**: “Although, the limited amount of data from this study does not allow us to conclusively identify the driving force responsible for the variation in ASVs of a given species,”.

Question: L547: “data” are plural.

Answer:

We have made the corresponding change in line 533: “
“Although, the limited data from this study do not allow us to conclusively identify the driving force responsible for the variation in ASVs of a given species,.....”

Question: 551-553: Why is it surprising/”striking” that are more common ASVs within species? Presence of common and rare genotypes and species is a common characteristic of biological populations and communities. Why do you assume a more common ASV is more functionally relevant and what specific function are you referring to?

Answer:

We have changed and clarified it in the revised version in **lines 543-544**:
“In addition, certain ASVs were detected more frequently within a predominant species than others. It is of great interest and importance to ascertain their relevance to disease potential.”.

Question: L561-563: Please clarify this sentence.

Answer:

In our point of view, the relevance/importance of a species in terms of disease potential should be assessed with frequency in combination with other factors, such as pathogenicity. We clarified this point in **lines 552-554**: "It is reasonable to assume that the dominance of single genotypes, represented in our study by dominant ASVs, has a selective advantage within a genus or species, for example, by increasing their pathogenicity and virulence on their hosts."

Question: L563-565: Please clarify this sentence. Are you referring to dominance of a genotype/ASV within species, not "genus"?

Answer:

Both could be true. A dominant genotype can have a selective advantage within the genus or among different genotypes of the same species. We clarified this in **lines 552-554**: "has a selective advantage within species or genus,".

Question: L566-575: As written, this is describing what a systematics study would focus on. Is this content important to the study? If it is included, it may be valuable to try to link the phylogenetic relationships among the common species to their ecology.

Answer:

You are right. Since the tree is created based on the ASVs, its value is limited. We have removed it in the revised version. This change meets also the requirements raised by other reviewers.

Question: L587-589: Please consider revising this sentence to improve clarity. In particular, it is unclear what "is likely varied, with implications for food security worldwide" means. Are you suggesting that climate change will alter the distribution of different *Pythium* spp. in soil?

Answer: We rewritten this **in lines 567-572**: "Further experiments are needed to determine the factors (e.g. temperature) that affect the virulence of *P. atrantheridium* and *P. ultimum* corn. As soil-borne pathogens are generally much better able to adapt to climate change, it can be assumed that the distribution and relative abundance of these pathogens including *Pythium* spp. are expected to increase, which will have an impact on global food security in the future"⁷⁹".

Question: L589-591: Please clarify how "occurrence" and "distribution" differ in this context.

Answer: We prefer "distribution". We deleted this sentence in the revised version."

Question: L598: Should "antagonistically" read "antagonistic"?

Answer:

Yes, we have made the correction in **line 580**: "SA- and JA-mediated signaling functions are antagonistic and often exploited by pathogens to promote their virulence"³⁴".

Question: L610: Please change "is" to "are" and "its" to "their" since *Pythium* spp. is a plural term.

Answer: We have deleted this part in the revised version.

Reviewer #3 (Remarks to the Author):

Main remark:

We have added more experimental details and descriptions into the M&M, as suggested. A point-by-point response/amendment can be found in the revised manuscript, which is highlighted in yellow.

I have several concerns, suggestions and edits that can be found in the attached pdf.

Answer: Thank you very much for your great efforts to help improving the manuscript.

• *More details are required in the Materials and Methods. I would not be able to repeat your study.*

Answer: We have added more details in the M&M.

• *Data are only different if they are statistically different ($P < 0.05$). Throughout the results, there are instances of "increasing tendency", "varied to some extent", "tended to be", "a slight difference", etc. Science is very exact. Either it increased or it didn't based on what the statistics detect.*

Answer: We fully agreed. We have added related information and tried to describe the data in an exact form throughout the manuscript.

- *You collected soil samples, presumably for DNA extraction, and also baited Pythium species from the soil using falcon tubes containing selective medium. In the results, it is not clear to me, if the number of Pythium species you report are from the soil samples-DNA sequencing or the soil baiting-isolation or both (summed together). Along this line, were differences between the Pythium species detected by sequencing and the Pythium species recovered by baiting. This would be very interesting information.*

Answer: We have added related information and details in the revised version. You can find them in our answers below.

Below are some line-by-line comments. Please also refer to the attached pdf.

Question: Abstract: Needs to be rewritten. I think you could include more details

Answer:

Yes, abstract has be rewritten.

Question: is plural therefore please edit the first sentence. Pythium spp. pose a serious threat to agricultural production worldwide, but their severity is often neglected because little knowledge about them are available.

Answer:

Thank you for the suggestion. We have used **in lines 35-36:** "Pythium spp. pose a serious threat to agricultural production worldwide, but their severity is often neglected because little knowledge about them is available."

Question: Line 37 delete "As a result,"

Answer:

We have deleted it in the revised version.

Question: Line 38-40 Please edit sentence for clarification. I think you could include more details, for example, range of the numbers of species recovered in each country. You could also include that 10 species were common across all countries.

Answer:

We have edited it and also added some details, but we are limited to 200 words. **In lines 38-41** we added "We also identified 82 species, with up to 21 species in a single soil sample, and the prevalent species, characterized by high species diversity, varying disease potential, and widespread in most countries."

Question: Line 41-42 I disagree that the Pythium species differ "Strongly". See my comments below.

Answer:

We rewritten this sentence in **lines 43-45:** "Infection experiments with recovered isolates indicate that Pythium species differed in their disease potential."

Question: Lines 43-45 The gene expression data could be a different sentence. Also, why only mention JA defenses when the relative change in expression of PR1 was 20-100 fold higher than the non-inoculated control?

Answer:

We added it in **lines 43-45:** Infection experiments with recovered isolates showed that *Pythium* spp. differed in disease potential, and that effective interference with plant hormone networks by upregulation of SA with simultaneous suppression of JA-mediated defenses is an essential component of the virulence mechanism of *Pythium* species."

Question: Line 47-48 I disagree that your work contributes to our understanding of the evolution of virulence and pathogenicity. And I have to question if it contributes to plant-pathogen interactions,

since the species you recovered were present in the soil, and no data are presented on species that were present in corn plants.

Answer:

The data of this study (a larger number of Pythium species including several known plant pathogens, high species diversity and varying disease potential) are a valuable metabarcoding dataset for further studies on Pythium.

It is also true that this study does not yet directly contribute to the understanding of host-pathogen interactions. But, isolates recovered from soil allow us for a comparative and systemic study on host-pathogen interactions at the molecular depth in the future.

We highlighted these points in **lines 46-50**: “This study provides a comprehensive and valuable metabarcoding dataset that enables deep insights into the species diversity of Pythium communities in European corn fields and the basis for a better understanding of plant-Pythium interactions, facilitating the development of an effective strategy to cope with this pathogen in the future.”.

Introduction

See pdf for editorial suggestions

Question: Line 67 Please use virulence rather than pathogenicity. “moderate pathogenicity” – according to APS, pathogenicity is the ability to cause disease. An organism can either cause disease or not. Virulence is a degree or measure of pathogenicity; the relative capacity to cause disease.

Note that Matthiessen et al. 2016 showed P. sylvaticum and other species had high virulence on corn.

Answer:

We have made the corresponding change in **lines 64-66**: “For instance, *P. ultimum* is particularly aggressive with a broad host range, while *P. sylvaticum* is a highly pathogenic species for soybean but has moderate virulence in corn as observed in the US⁸.”.

Question: Line 70-72 This sentence seems out of place. And perhaps should be included in the discussion with reference to the pathogenicity tests you did.

Answer:

We would like to keep the sentence in the introduction as it is part of the currently limited information about the corn-Pythium interactions.

Question: Line 110-111 how about SA?

Answer: We have included SA lines **108-110**: “We show that effective interference with plant hormone networks to upregulate SA-signalling while simultaneously suppressing JA-mediated defences may be a crucial part of the virulence mechanism of pathogenic *Pythium* spp..”.

Materials and Methods

Question: Significant details were missing. I am not sure I would be able to repeat your study. See pdf for editorial suggestions and additional comments.

Answer:

We have added more details in the M&M, as the reviewer suggested, in **lines 116-126**, in which information about “bait” sampling and species cultivation are also included.

Question: Line 116-130 Soil samples were collected 4 weeks after corn seeding and placed in a 50 ml tube – how did you collect the soil samples? Shovel/soil probe? How many samples were collected from each site to place in the 50 ml tube? Did you collect the soil samples from the plant row? Did you mix the soil and then place a sample of it in the 50 ml tube? What was this soil used for? I presume DNA extraction but it is not clear from your methods.

Answer:

We have made corresponding changes and added more details in **lines 117-119**: “For each site, eight 50 ml soil samples were taken by pressing the 50 ml falcons (Sarstedt, Nürnberg, Germany) into the soil with the opening facing downwards. Soil samples were stored at -20 °C.....”.

Question: Line 121-123 what do you mean by “dominantly characterized by intensive cultivation of corn”? is corn grown in those fields every year? Is it rotated with something? Table S12 (Should be Table S1, since it is the first supplementary table presented) does not have precrop (do you mean previous crop) or tillage data. I agree these may be helpful data to include so if you do have those data I’d like to see them in the revision.

Answer:

We changed the order of the Tables. We changed the sentence in **lines 116-119**: “Sampling sites were selected, where corn was grown in all the years.”. Corn has been grown very frequently at these sampling sites. However, this does not mean that only corn was grown there in monoculture, as this is not permitted in Europe.”.

Due to the complexity and in particular the incompleteness of the information on "previous crops" and "tillage" at the individual sampling sites, we decided to delete these in the revised version (Table S1).

Question: Line 123-129 When were the baits placed in the soil? So, they were 15cm from the plant row. This is quite a distance from the corn seedlings, so they were likely sampling Pythium species present in the bulk soil, rather than the rhizosphere soil. How were the baits spaced apart from each other? How many holes were in each bait tube? Where were the holes positioned on the tube – at different heights? Different sides?

Answer:

We have added the corresponding information in **lines 119-126**: “For „bait“ sampling and species cultivation, Benomyl-Ampicillin-Rifampicin-Pentachloronitrobenzol was (Table S1) filled into 15 ml centrifuge tubes (Sarstedt, Nürnberg, Germany). After solidification of the medium, 1 mm holes were punctured into the tubes using a hot needle. Eight baits were placed in the soil of a sampling site on the day of seeding. Before inserting the tube into the soil, a hole was drilled with an empty tube with 15 cm apart from the corn row. After four weeks, the baits were collected and stored at 4 °C until further processing. An overview of sampling sites is provided in Table S2.”.

Question: Lines 141-169 Of all the species, why did you choose to evaluate the pathogenicity of P. attrantheridium and P. ultimum? How many isolates of each species did you evaluate? Please use ‘inoculated’ rather than ‘infected’ – you inoculated corn with the pathogen, and then evaluated infection. Inoculation does not always cause infection. What was the ratio of Pythium-colonized millet:soil substrate? I do not understand this sentence “Because of minimal soil residues, an exact measurement of root fresh weight was not possible.”

Answer:

Yes, *P. ultimum* and *P. attrantheridium*, are two common species in Europe. And, the both isolates we used were confirmed by comparing the Sanger sequencing with the dominant ASVs detected from corresponding species by NGS survey. Since the diversity of these two species were dominated by single ASVs, they were defined as dominant genotypes und used for further analysis. We added this in M&M, **lines 193-195**: Dominant genotypes were determined by comparing the Sanger sequencing data with the dominant ASVs of corresponding species identified by NGS survey, and used for further experiments.” And, in **lines 222-224**: “Mycelial plugs of the isolates *P. ultimum* and *P. attrantheridium*, which are widespread in Europe, were placed adjacent to the roots for infection.”. In addition, we added more details about the ratio of Pythium-Millet and soil substrate in **lines 199-202**.

It is difficult to measure the fresh weight of the roots accurately, as soil residue always remains attached to the roots. Even minimal soil residue distorts the fresh weight of the roots considerably. For this reason, we decided to wash the roots and then measure the root dry weight instead of the root fresh weight. We have clarified this in **lines 210-211**. “Shoot fresh and root dry weight (at 65 °C for 2 days) were determined”.

Question: Line 159-168 I am having a hard time visualizing what you did here. Why were the corn seeds gown under short day conditions?

Answer:

Corn is a short-day plant. Generative growth begins when the day falls below a certain length. We clarified this point in **lines 208-210, and 215-216**.

Question: Line 170-171 Unclear. Were the 8 individual soil samples all in 8 individual 50 ml tubes, that were mixed and then a 5 g subsample removed? If yes, this information should be in the soil sampling section. In this line suggest you rewrite the sentence "For gDNA isolation, a 5g soil sample was removed from the 50ml Falcon tube sample of soil collected from each sampled field".

Answer:

The 8 individual soil samples were in 8 separate 50 ml tubes. A pooled 5 g sample was prepared from the 8 individual samples in the laboratory under sterile conditions. We clarified this point in **lines 128-129**: "For gDNA isolation from soil, a pooled 5 g soil sample was prepared by mixing the 8 individual soil samples from each sample site."

Question: Line 180-181 Not sure what you mean "was used justified by and a". How did you "optimize" the primer pairs? Did you design the primer pairs? If not, please provide a reference.

Answer:

Yes, the primers were specially designed for this study and were tested for their sensitivity and specificity to *Pythium*. We clarified this in **lines 140-146**: "For this study, the ITS1 region was amplified using the primer pair pyth_f (5'-3'; TCGGGAAGGATCATTACCACAC) and pyth_r2 (5'-3'; GCGTTCAAATTCGATGACTC), which were specifically designed based on a multiple sequence alignment of all available ITS sequences of *Pythium* spp. (NCBI December 2022). Fungal sequences were included to focus on regions offering high dissimilarity. Further, primer specificity and efficiency were finetuned by qPCR with *Pythium* isolates."

Question: Line 186 "quantity and quality" of what?

Answer:

We clarified this in **lines 147-149**: "The DNA quantity and quality were measured by photometric measurement using the "Thermo Scientific Multiskan GO" (Thermo Fisher, Massachusetts, USA) and determined by real-time PCR."

Question: Line 207 If there are 171 Pythium species they must all be different. "different" is redundant – please delete here and throughout the manuscript.

Answer:

We have removed it throughout the manuscript.

Question: Line 218-222 More details are required on how you prepared the plants, and inoculated them for RT-q-PCR analysis. Please see pdf.

Answer:

We have added more details as suggested by the reviewer in M&M, **lines 222-224**.

Question: Line 231 Please include the name of the authors and then the superscript for the reference, e.g. Pfaffel⁵². There are several cases of this in the discussion too.

Answer:

Yes, we'd like, but unfortunately, we have no influence on this point, as this type of citation is a requirement of the journal.

Results

Question: In the results, usually only observations are reported. Throughout this section there are sentences that pertain to the Materials and Methods, or Discussion. Please check the attached pdf.

Answer:

We have moved and or removed some of such sentences, accordingly.

Question: Line 283-284 This sentence should be in the Discussion

Answer: Definitely! We have made it.

Question: Line 285 & 287 What is the difference between “sampled sites” and “sampled locations”? Be consistent with terminology.

Answer:

Many thanks. We prefer to use “sampled sites” and adjusted this e.g. in **line 285**.

Question: Line 284 Delete “Additionally,”

Answer:

It has been deleted.

Line 292-300 Lots of redundancy, I have made some suggestions to the text.

Answer:

Thank you very much. As you suggested, we changed this part, accordingly, lines 293-304

Question: Line 303-312 I disagree with your statements. There are no statistical differences among years for these species, and therefore there was no difference in relative abundance for these species among years. Please rewrite.

Answer:

We have rewritten and simplified this part, **lines 305-310**.

Question: Figure 2 Please insert years, move the taxa legend to the far left since that is the first heat map we see. Figure 3 No significant differences () are indicated. I inserted them from Table S3. Please check. What method did you use to detect statistical differences? LSD? Tukeys? Please include this information in the Methods text as well.*

Answer:

Yes, we have made corresponding changes as suggested.

The multiple comparisons were performed using an appropriate user-defined contrast for the corresponding influencing variables with the multcomp package in R. as described by “Hothorn, T., Bretz, F. & Westfall, P. (2008) Simultaneous Inference in General Parametric Models. 789 Biom. J. 50, 346-363; 10.1002/bimj.200810425”.

We have inserted the significant asterisk. We have also included the figure caption: Statistical differences were analyzed by a multiple contrast test for *** $p < 0.001$, ** $p < 0.01$, * $p < 0.05$.

The information regarding the method for detect statistical differences is available in the section “Statistical analysis” in M&M, in **lines 245-260**.

Question: Line 325 No idea what you mean by “latent”

Answer:

Yes, it is not the correct term. We have changed this sentence in **lines 321-323**: “ However, the abundance of *P. heterothallicum*, *P. sp.*, and *P. aristosporum* were very low in the Netherlands during the three years.”.

Question: Line 327-350 Since soil texture did not affect Pythium inoculum density, these sentences need to be deleted. I have suggested a sentence that can be used on the attached pdf.

Answer:

We have rewritten this session according to the re-analysed data in lines 327-341.

Question: Line 350-352 please clarify.

Answer:

As this sentence has no essential meaning, we have deleted it (see the point above).

Question: Line 359 “higher diversity” – where is the statistics to prove this?

Answer:

Unfortunately, statistical proof is not possible, as this is only a single added value and there is no variance.

Question: Line 371 “difference is ASV counts” – statistics?

Answer:

For this difference, statistical prove is also not possible, as this is only a single added value, so there is no variance.

Question: Figure 5 I wonder if the data for All years could be shown as a bar graph. It might be easier to compare among species?

Answer:

Yes, we added the number of ASVs as text label in Figure 6.

Question: Line 376, 379-380 please clarify.

Answer:

We have clarified this by replacing distribution with occurrence in **line 362**: “we analyzed their corresponding occurrence.”, and in **lines 364-365**: “for *P. attrantheridium*, *P. oligandrum*, *P. rostratifyingens*, *P. sylvaticum*, and *P. ultimum*, only one ASV was detected from almost all sampled sites.”.

Question: Figure 6B and line 390-402 It is not clear to me what the purpose of the phylogenetic analysis was in your study. Did you observe different phylogenetic relationships to what has been reported before among Pythium species?

Answer:

Agreed! We have replaced this part with a co-occurrence between *Pythium* spp. and their linkage with soil parameters in order to provide a ecological respective as suggested by you and the reviewer II (see lines **375-391**).

Question: Line 410-411 did you not assess root rot severity? Fig. 7C, D and E are okay but a bar chart of root rot with SE bars and statistics support your statement “more severe disease symptoms”. Also, these are stereomicroscopic images, right?

Answer:

Yes, these are stereomicroscopic images. We did not assess the severity of root rot either directly or indirectly. Nevertheless, the images show disease symptoms to varying degrees, which supports the measured reduction in shoot fresh mass and root dry mass caused by the infection. We clarified this in lines **396-401**: “Both species caused characteristic disease symptoms, but differed in the disease's severity. Thereby, *P. ultimum* showed more pronounced disease symptoms of root rot compared to *P. attrantheridium* (Figure 10E and G). The germination rate of corn was severely affected by infection with *P. ultimum*, but not with *P. attrantheridium* (Figure 10A, Table S11).”.

*Question: Line 410-415 I like to include the exact P-value (to 2-3 decimal places) when I am stating differences. I am not sure if Nature Communications likes that so it would eb worth checking some recent publications to see if exact P-values are used. If yes, then please use them throughout the results. Figure 8 Spell out everything in the Figure legend. *** = P<0.001? – please include in legend.*

Answer:

The exact p-value is given in the corresponding table in the appendix throughout the manuscript. In line with your suggestion, we have added the change in the legend of Figure 10: “Statistical differences were tested by a multiple contrast test for *** $p < 0.001$, ** $p < 0.01$, * $p < 0.05$.”

Discussion

I like this link to how to write a discussion – particularly the first paragraph

<https://www.biosciencewriters.com/How-to-Write-a-Strong-Discussion-in-Scientific-Manuscripts.aspx>

Answer:

Thank you very much, we got it!

Question: Line 440-442 belongs in results “Pythium was detected in the soil at 125 of 127 locations.

Answer:

We have removed this sentence and rewritten this section in **lines 414-427.**

Question: Line 438-477 need to be reworked. The discussion goes back and forth between the DNA sequencing and isolation on selective media and consequently is difficult to follow. If you include data on the Pythium species detected by sequencing, and the Pythium species recovered by baiting, this would help with your discussion and the points you make.

Answer:

We have included the bait data in the results section. Following your suggestion, we also have rewritten this part of the discussion in lines **414-427.**

Question: Line 463 not sure what you mean by “scattering”

Answer:

The Shannon-Wiener Index, a mathematical quantity, was used to describe diversity. It describes the diversity in the samples considering both the number of different taxa and the abundance. Morita & Akao (2021) reported that the Shannon-Wiener index was different for DNA isolations with 5 g of soil from smaller amounts of soil. Also following the suggestion from the reviewers, we removed this “technical issue” from the discussion.

Question: Line 469-470 how did you optimize the primers? Why did they reduce fungal amplicons?

Answer:

We have clarified this in both M&M in **lines 141-146:** “For this study, the ITS1 region was amplified using the primer pair pyth_f (5’-3’; TCGGGAAGGATCATTACCACAC) and pyth_r2 (5’-3’; GCGTTCAAATTTGATGACTC), which were specifically designed based on a multiple sequence alignment of all available ITS sequences of *Pythium* spp. (NCBI database Version??). Fungal sequences were included to focus on regions offering high dissimilarity. Further, primer specificity and efficiency were finetuned by qPCR with *Pythium* isolates.”, and in the discussion in **lines 429-431.** “Application of *Pythium* specific primers greatly increased the efficiency and specificity of the ITS-PCR amplification as well as the taxonomic resolution of the data. The primers specifically amplified the ITS1 region of *Pythium* and reduced fungal amplicons.”.

Question: Line 478 This is not genetic diversity – the paragraphs below discuss species diversity.

Answer:

We have changed it in **line 475:** “High species diversity of *Pythium* communities in Europe” and also throughout the manuscript.

Question: Line 489 Broders et al 2007, Jaing et al 2012, Matthiesen et al 2016, Rojas et al. 2019 reported Pythium spp. diversity in corn roots so I am curious why you chose to compare your data to Rojas et al. 2017 which describes Pythium spp. recovered from soybean? I would encourage you to do a more thorough discussion of Pythium spp. that have been reported from corn. Bickel and Koehler 2021 have a good review.

Answer:

Yes, we have referred to the studies you mentioned in the previous section. We tried to make it clear that some species identified from this study have a dominant role, globally. We thank you for your suggestion and have slightly reworded the section, **lines 484-502.**

Question: Line 512 what do you mean by 'particularly pronounced'.

Answer:

We have replaced it with "higher" in **lines 507-508**: "It has been reported⁷¹ that spring precipitation totals were particularly higher in these two countries."

Question: Line 515-516 – This needs to be expanded on. Moreover, this is a shortcoming of your study: you have reported Pythium species present in the bulk soil of corn fields; it is possible there is less diversity of Pythium species in the rhizosphere, and within symptomatic corn rots.

Answer:

A critical but important question. It is generally believed that the diversity is lower in symptomatic corn roots, which may partially explain the differences between our study and studies that mainly focused on infected corn roots. It is true that the life form of *Pythium* in the soil differs from that in the rhizosphere. Whether or how the rhizosphere influences "species diversity" is a question that needs to be investigated in the future, but is beyond the scope of this study. We clarified it in **lines 509-514**: "Another possible scenario for the minor differences among countries could be that the results of the bulk soil samples rather reflect a natural, stable state of the *Pythium* community that has been less influenced by e.g. infection pressure. This could also explain some discrepancies between our data and other studies which mainly focused on infected roots. Therefore, it is of great interest to ascertain the diversity in the rhizosphere and within the symptomatic corn rots."

Question: Line 520 You did not report data for 'agricultural history'.

Answer:

No, we didn't. We tried to obtain the 'agricultural history' data from the different sampling sites. The problem is that the data are very complex and most of them are not complete. Nevertheless, our preliminary analyses showed no significant impact on the diversity of *Pythium* spp. This result is somehow to be expected as *Pythium* is able to persist in the soil for many years by forming permanent spores (oospores). Therefore, it would have been necessary to document e.g. the crop rotation of the last 5-10 years to really assess the effects. Unfortunately, such data sets were not yet available for this project.

For this reason, we did not include the data in the manuscript in order to avoid ambiguity. We changed in **lines 517-519**: "Nevertheless, our data show an more even distribution of the predominant *Pythium* species in 11 European countries, despite different geographical locations and cropping systems, thus indicating a high infection potential of *Pythium* in Europe."

Question: Line 521 Significant differences among soil texture classes were only detected for number of species detected (or recovered).

Answer:

Thank you for the suggestion. We have changed it accordingly in **line 519**.

Question: Line 522 Delete. Inoculum density was not significant for soil texture.

Answer:

We have deleted in the revised version.

Question: Line 533-538 There are no data to support soil suppressiveness. Delete.

Answer:

Agreed. It has been deleted in the revised version

Question: Line 543-546 This needs further explanation. What are the rationales for proposing this?

Answer: We have changed it in **lines 535-539**: "In particular, *P. rostratiformis*, *P. sylvaticum* and *P. ultimum* were found to have a higher number of different ASVs in 2020. Although, the limited data from this study do not allow us to conclusively identify the driving force responsible for the variation in ASVs of a given species, soil moisture seems to be a critical role in this process, as prerequisite for zoospore movement in soil⁷⁷".

Question: Line 566-575 I am not familiar with Pythium phylogeny and I suspect most of your readers will not be too. Is this any different than the relationships that have been described before? Again, I don't think these data add anything to your study. I would far rather see a comparison of the Pythium species detected in the soil by DNA sequencing or recovered by baiting.

Answer:

An absolute justified question. As explained, the phylogenetic analysis and the tree do not provide a meaningful message, so we have removed the phylogeny in the revised version. This change is also in line with the requests of other reviewers.

Question: Line 582-586 I think this statement is overreaching. Who else has reported P. attrantheridium as a pathogen of corn? Is it possible that this species could cause more severe disease at lower temperatures compared to what you tested it at (20o C)? Rojas et al and Mathhiesen et al both report species that are high virulent at 13 o C and significantly less so at 18 o C and/or 23 o C.

Answer:

Broders et al. (2007) have made the first report of *P. attrantheridium* as a pathogen of both corn and soybean seedlings⁸. Although we did not have the data from different temperatures, Rojas et al. (2019) investigated the weight of the plant as a function of temperature, and showed that the difference to the non-infected control was greater at a temperature of 20 °C than at 15 °C. This could be attributed to slower plant growth at lower temperatures. Matthiesen et al. (2016) have also reported that the root mass was higher at 18 °C and 23 °C than at 13 °C, which consequently affected the calculation of the percentage of root rot (reduces)¹². Therefore, they calculated the seed rot severity in relation to root growth. For this reason, we decided to choose a temperature of 20 °C for our infection experiments. We clarified this point in **lines 567-572**: "Further experiments are needed to determine the factors (e.g. temperature) that affect the virulence of *P. attrantheridium* and *P. ultimum* on corn."

Question: Line 588-589 Please clarify. What are the rationales for suggesting this?

Answer:

This is based on two main aspects: limited/changing land use worldwide and the fact that soil-borne pathogens generally much better able to adapt to climate change (Delgado-Baquerizo et al., 2020, John T. Bickel and Alyssa M. Koehler 2021). We clarified this **in lines 569-572**: "As soil-borne pathogens are generally much better able to adapt to climate change, it can be assumed that the distribution and relative abundance of these pathogens including *Pythium* spp. are expected to increase, which will have an impact on global food security in the future⁷⁹."

Question: Line 592-608 I'd like to see more of a discussion regarding the strong expression of PR1.

Answer:

Significance of the crosstalk between SA and JA-mediated signalling in plant defence response is well documented as mentioned in the discussion. We found that both *P. ultimum* and *P. attrantheridium* were able to strongly upregulate *PR1* while down-regulated *PDF1.2*. Thus, an enhanced SA- and simultaneously suppressed JA-signaling pathway seems belonging to the virulence strategy. We have rewritten this session in lines **573-592** to highlight this point.

Question: Line 615 "with great disease potential" - based on what data?

Answer:

We have deleted this part in the revised version to avoid redundancy

Supplementary materials

Question: Table S1. Was this protocol completely developed by you or did you modify an existing protocol? If the latter, the existing protocol should be referenced. Check spelling of ammoniumbromid – think it ends in 'e'

Answer:

It has been changed. We indicate this also in **lines 139-131**: “The gDNA Isolation was performed using a modified CTAB as described by³⁷ (**Table S3**).”.

Question: Table S2. Delete “Primers used in this study.” Redundant with next sentence. Do these need to be referenced?

Answer:

It has been changed (**Table S5**).

*Question: Table S3 (and all other pertinent tables). Please indicate what o, *, ** and *** are? Presume P<0.1, 0.05, 0.01 and 0.001 respectively? And what statistical test you used.*

Answer:

We clarified these in the revised version (**Table S6**). “Statistical comparison of the three sampling years 2019, 2020 and 2021 of the 10 most abundant *Pythium* species, *P. aristosporum*, *P. attrantheridium*, *P. heterothallicum*, *P. intermedium*, *P. monospermum*, *P. oligandrum*, *P. rostratiformis*, *P. sp.*, *P. sylvaticum* and *P. ultimum*. Statistical differences were tested two sided by a multiple contrast test for *** $p < 0.001$, ** $p < 0.01$, * $p < 0.05$.”.

Question: Table S4. Netherlands - insert ‘s’

Answer:

It has been changed in **Table S7**: “Statistical comparison of the 10 sampled countries Austria, Belgium, France, Germany, Hungary, Italy, Netherlands, Romania, Spain, Switzerland, with the grand Mean of all countries pooled for the 10 most abundant *Pythium* species, *P. aristosporum*, *P. attrantheridium*, *P. heterothallicum*, *P. intermedium*, *P. monospermum*, *P. oligandrum*, *P. rostratiformis*, *P. sp.*, *P. sylvaticum* and *P. ultimum*. Statistically differences compared to the grand Mean were tested two-sided by a multiple contrast for *** $p < 0.001$, ** $p < 0.01$, * $p < 0.05$.”.

Question: Table S5. Edit last phrase of heading: (designation of the soil type according to ONORM L 105080). Insert e in Soiltyp* heading of the table.*

Answer:

It has been changed in **Table S8**. “Classification of soil texture class according to clay content or soil type. S = sand, U = silt, T = clay, L = loam, s = sandy, u = silty, t = clay, l = loamy (* designation of the soil type according to ÖNORM L 1050).”.

Question: Table S6. Modify title “Abundance of Pythium estimates based on soil texture classes (heavy, medium and light) compared to the mean of Pythium abundance for all sites sampled.”

Answer:

It has been changed in **Table S9**: “Abundance of *Pythium* estimates based on soil texture classes (heavy, medium and light) compared to the mean of *Pythium* abundance for all sites sampled. Significant differences were tested against the mean of all samples by a multiple contrast for *** $p < 0.001$, ** $p < 0.01$, * $p < 0.05$.”.

Question: Table S7. Modify title “Estimates of different Pythium species based on soil texture classes (heavy, medium and light) compared to the mean number of Pythium species for all sites sampled.”

Answer:

It has been modified in **Table S10**: “Estimates of *Pythium* species based on soil texture classes (heavy, medium and light) compared to the mean number of *Pythium* species for all sites sampled. Significant differences were tested against the mean of all samples by a multiple contrast for *** $p < 0.001$, ** $p < 0.01$, * $p < 0.05$.”.

*Question: Table S8. Modify title “Comparison of the germination rate of corn inoculated with Pythium attrantheridium and P. ultimum with a non-inoculated control. * indicates significance at P<0.05 based on what statistical test?*

Answer:

It has been modified in **Table S11**: “Comparison of the germination rate of corn inoculated with *P. attrantheridium* and *P. ultimum* with a non-inoculated control. Statistical differences were tested two sided by a multiple contrast test for *** $p < 0.001$, ** $p < 0.01$, * $p < 0.05$.”.

*Question: Table S9. Modify title “Comparison of the shoot fresh weight per plant of corn inoculated with *P. attrantheridium* and *P. ultimum* with a non-inoculated control. * indicates significance at $P < 0.05$ based on what statistical test?*

Answer:

It has been modified in **Table S12**: “Comparison of the shoot fresh weight per plant of corn inoculated with *P. attrantheridium* and *P. ultimum* with a non-inoculated control. Statistical differences were tested two sided by a multiple contrast test for *** $p < 0.001$, ** $p < 0.01$, * $p < 0.05$.”.

*Question: Table S10. Modify title “Comparison of the root dry weight per plant of corn inoculated with *Pythium attrantheridium* and *P. ultimum* with a non-inoculated control. * indicates significance at $P < 0.05$ based on what statistical test?*

Answer:

It has been modified in **Table S13**: “Comparison of the root dry weight per plant of corn inoculated with *P. attrantheridium* and *P. ultimum* with a non-inoculated control (NC). Statistical differences were tested two sided by a multiple contrast test for *** $p < 0.001$, ** $p < 0.01$, * $p < 0.05$.”.

*Question: Table S11. Modify title “Comparison of gene expression of PR1, PDF1.2, JA, ETR2 and NCED3 in corn seedlings inoculated with *Pythium attrantheridium* (P2), *P. ultimum* (P38) or non-inoculated (NC). *** indicates differences between expression levels at $P < 0.001$ using a two sided xxx test.*

Answer:

It has been modified in Table S14: “Comparison of gene expression of PR1, PDF1.2, JA, ETR2 and NCED3 in corn seedlings inoculated with *P. attrantheridium* (P2), *P. ultimum* (P38) or non-inoculated (NC). Statistical differences were tested two sided by a multiple contrast test for *** $p < 0.001$, ** $p < 0.01$, * $p < 0.05$.”.

Table S13. Was this protocol completely developed by you or did you modify an existing protocol? If the latter, the existing protocol should be referenced.

Answer:

The protocol was developed by Syngenta Agro GmbH (Frankfurt am Main, Germany). It is indicated in **Table S1**.

References

- Broders, K. D., Lipps, P. E., Paul, P. A., & Dorrance, A. E. (2007). Characterization of *Pythium* spp. Associated with Corn and Soybean Seed and Seedling Disease in Ohio. *Plant Disease*, 91(6), 727–735. <https://doi.org/10.1094/PDIS-91-6-0727>
- Delgado-Baquerizo, M., Guerra, C. A., Cano-Díaz, C., Egidi, E., Wang, J.-T., Eisenhauer, N., . . . Maestre, F. T. (2020). The proportion of soil-borne pathogens increases with warming at the global scale. *Nature Climate Change*, 10(6), 550–554. <https://doi.org/10.1038/s41558-020-0759-3>
- Delgado-Baquerizo, M., Maestre, F. T., Reich, P. B., Jeffries, T. C., Gaitan, J. J., Encinar, D., . . . Singh, B. K. (2016). Microbial diversity drives multifunctionality in terrestrial ecosystems. *Nature Communications*, 7, 10541. <https://doi.org/10.1038/ncomms10541>
- Hershkovitz, M. A., & Lewis, L. A. (1996). Deep-level diagnostic value of the rDNA-ITS region. *Molecular biology and evolution*, 13(9), 1276–1295. <https://doi.org/10.1093/oxfordjournals.molbev.a025693>
- Hothorn, T., Bretz, F., & Westfall, P. (2008). Simultaneous Inference in General Parametric Models. *Biometrical Journal*, 50(3), 346–363. <https://doi.org/10.1002/bimj.200810425>

- Matthiesen, R. L., Ahmad, A. A., & Robertson, A. E. (2016). Temperature Affects Aggressiveness and Fungicide Sensitivity of Four *Pythium* spp. that Cause Soybean and Corn Damping Off in Iowa. *Plant Disease*, *100*(3), 583–591. <https://doi.org/10.1094/PDIS-04-15-0487-RE>
- Morita, H., & Akao, S. (2021). The effect of soil sample size, for practical DNA extraction, on soil microbial diversity in different taxonomic ranks. *PLoS One*, *16*(11), e0260121. <https://doi.org/10.1371/journal.pone.0260121>
- Pfaffl, M. W. (2001). A new mathematical model for relative quantification in real-time RT-PCR. *Nucleic Acids Research*, *29*(9), e45. <https://doi.org/10.1093/nar/29.9.e45>
- Robideau, G. P., Cock, A. W. A. M. de, Coffey, M. D., Voglmayr, H., Brouwer, H., Bala, K., . . . Lévesque, C. A. (2011). DNA barcoding of oomycetes with cytochrome c oxidase subunit I and internal transcribed spacer. *Molecular Ecology Resources*, *11*(6), 1002–1011. <https://doi.org/10.1111/j.1755-0998.2011.03041.x>
- Rogers, S. O., & Bendich, A. J. (1985). Extraction of DNA from milligram amounts of fresh, herbarium and mummified plant tissues. *Plant Molecular Biology*, *5*(2), 69–76. <https://doi.org/10.1007/bf00020088>
- Rojas, J. A., Witte, A., Noel, Z. A., Jacobs, J. L., & Chilvers, M. I. (2019). Diversity and Characterization of Oomycetes Associated with Corn Seedlings in Michigan. *Phytobiomes Journal*, *3*(3), 224–234. <https://doi.org/10.1094/PBIOMES-12-18-0059-R>
- Wielgoss, A. M. (2009). *Dynamik der schilfassozierten Oomycetengemeinschaft im Litoral des Bodensees unter besonderer Berücksichtigung des Schilfpathogens Pythium phragmitis*.

REVIEWER COMMENTS

Reviewer #1 (Remarks to the Author):

I appreciate the authors addressing my previous concerns, but there are still serious methodological and taxonomic issues that need to be addressed as they draw into questions all of the results. A more thorough review can not be completed until these are addressed. Specifically, these include:

1) The custom data base developed in L163-167 is inaccurate and should not be used. The authors use “A custom Pythium database was created based on NCBI Genbank entries (December 2022) using the esearch tool⁴⁵. All protist sequences referring to the term “Pythium internal transcribed spacer” were downloaded and properly formatted in preparation of downstream analysis using a custom python script. The final database consists of 11.431 sequences containing 8.342 Pythium and 3.075 Globisporangium sequences.” Genbank is well known for have many dubious and inaccurate identification in the database. A custom database should only include type specimens or vouchered/verified specimens. For most oomycete meta-barcoding studies the standard has been the ITS and COXI databases used by Robideau et al. 2011 <https://doi.org/10.1111/j.1755-0998.2011.03041.x>. The other option is the curated database available from UNITE <https://unite.ut.ee/>. By using the Genbank-based database you are likely mis-classifying many strain. You also have duplicated many species that have recently undergone a taxonomic name change, for example Pythium sylvaticum is now Globisporangium sylvaticum, but you have used both names in your custom database. There are many examples of this specific type of error in all of the figures. You simply can not trust the accuracy of taxonomic classification of sequence data in Genbank. You need to use a curated database of accurate oomycetes.

2) The Pythium – Globisporangium taxonomy issue. It is not clear to me the author understand the taxonomic history of Pythium. Pythium sensu latu, includes several now accepted genera including Globisporangium, Elongisporangium, Phytopythium and Pythium <https://doi.org/10.1080/00275514.2022.2045116>. 7 of the 10 most frequently overserved species are in fact Globisporangium. These taxonomic changes are relatively new, so both names occupy GenBank. This has resulted in your use of both G. sylvaticum and P. sylvaticum in your analyses and figures, but in fact they are the same organism. The same is true for G. heterothalicum/P. heterothalicum and G. rostratifingens/P. rostratifingens and G. violae/P. violaei, etc. The ASVs for each of these pairs of species should be combined. For the sake of simplicity, I am fine with the authors using the terminology Pythium sensu latu in the manuscript, but they do need to address the taxonomic history of Pythium and Globisporangium in the introduction. When developing figures, the authors should consider making a figure similar to Figure 3 in Rojas et al. 2017 <http://dx.doi.org/10.1094/PHYTO-04-16-0177-R>, which breaks down oomycete species by Pythium clade outlined in Robideau et al. 2011 and <https://doi.org/10.1080/00275514.2022.2045116>. This way you can continue to use Pythium sensu latu, but distinguish between the different clades/species in the Pythiales.

3) The number of ASVs associated with “Pythium sp.” You indicate there are 77 ASVs of Pythium sp. Are you sure that all of these ASVs represent a single phylogenetically distinct Pythium species or

do they represent multiple "P. sp". This needs to be clarified in the methods. How do you assign taxonomy to unknowns? Based on your methods your custom database would include all "Pythium sp" in Genbank, which may compose dozens of phylogenetically distinct species. There may be multiple phylogenetically distinct species in this group of 77 ASVs. Clarification is needed.

4) All figures are drawn into questions based on the dual use of Globisporangium and Pythium names for the same species. See notes in attached document.

5) All uses of "Pythium" should be italicized unless you are referring to the disease such as "Pythium root rot" or "Pythium seedling disease"

6) Additional comments and suggested edits are included in that attached document.

Reviewer #2 (Remarks to the Author):

The authors have addressed prior comments through revisions or appropriate rebuttals. There may still be minor editorial issues that need to be addressed.

Reviewer #3 (Remarks to the Author):

Thanks for revising your manuscript based on the suggestions by the reviewers. I hope you agree the manuscript is much improved. I am happy with the revisions you have made. This is an excellent study that will contribute greatly to our understanding of the diversity of Pythium species in Europe.

To reviewer 1:

Comment 1:

The custom data base developed in L163-167 is inaccurate and should not be used. The authors use “A custom Pythium database was created based on NCBI Genbank entries (December 2022) using the esearch tool⁴⁵. All protist sequences referring to the term “Pythium internal transcribed spacer” were downloaded and properly formatted in preparation of downstream analysis using a custom python script. The final database consists of 11.431 sequences containing 8.342 Pythium and 3.075 Globisporangium sequences.” Genbank is well known for have many dubious and inaccurate identification in the database. A custom database should only include type specimens or vouchered/verified specimens. For most oomycete meta-barcoding studies the standard has been the ITS and COXI databases used by Robideau et al. 2011 <https://doi.org/10.1111/j.1755-0998.2011.03041.x>. The other option is the curated database available from UNITE <https://unite.ut.ee/>. By using the Genbank-based database you are likely mis-classifying many strain. You also have duplicated many species that have recently undergone a taxonomic name change, for example Pythium sylvaticum is now Globisporangium sylvaticum, but you have used both names in your custom database. There are many examples of this specific type of error in all of the figures. You simply can not trust the accuracy of taxonomic classification of sequence data in Genbank. You need to use a curated database of accurate oomycetes.

Answer:

We thank the reviewer's valuable suggestion. Although this required a complete re-analysis of the data, and considerable bioinformatics efforts and resources, we decided to follow it and reanalyzed our data using the database of Robideau et al. (2011) under <https://doi.org/10.1111/j.1755-0998.2011.03041.x> and an updated version of the taxonomic classification of genera according to Nguyen et. al. (2022) under <https://doi.org/10.1080/00275514.-2022.2045116>.

We are very pleased and also encouraged that this reanalysis has not led to any substantial changes in the key results and claims of this paper compared with our previous version and thus provides additional support for this study. Nevertheless, it is worth noting that, as the reviewer argues, the reanalysis provides more accurate and precise information on the taxonomic classification and name, and clarifies several erroneous information and interpretations in our previous version, which were mainly caused by using the custom database. We have therefore consequently changed the manuscript in M&M, main text and all figures) with the re-analysis results. The main changes are yellow-highlight in the revised version (i.a. lines 167-172, lines 261-297, lines 315-320, lines 340-349, lines 360-375 and lines 413-426).

We feel that the re-analysis of our data makes in particular the taxonomic classification much clearer than before and thus significantly improves the manuscript.

Comment 2:

The *Pythium* – *Globisporangium* taxonomy issue. It is not clear to me the author understand the taxonomic history of *Pythium*. *Pythium sensu lato*, includes several now accepted genera including *Globisporangium*, *Elongisporangium*, *Phytopythium* and *Pythium* <https://doi.org/10.1080/00275514.2022.2045116>. 7 of the 10 most frequently overserved species are in fact *Globisporangium*. These taxonomic changes are relatively new, so both names occupy GenBank. This has resulted in your use of both *G. sylvaticum* and *P. sylvaticum* in your analyses and figures, but in fact they are the same organism. The same is true for *G. heterothallicum*/*P. heterothallicum* and *G. rostratifingens*/*P. rostratifingens* and *G. violae*/*P. violaei*, etc. The ASVs for each of these pairs of species should be combined. For the sake of simplicity, I am fine with the authors using the terminology *Pythium sensu lato* in the manuscript, but they do need to address the taxonomic history of *Pythium* and *Globisporangium* in the introduction. When developing figures, the authors should consider making a figure similar to Figure 3 in Rojas et al. 2017 <http://dx.doi.org/10.1094/PHYTO-04-16-0177-R>, which breaks down oomycete species by *Pythium* clade outlined in Robideau et al. 2011 and <https://doi.org/10.1080/00275514.2022.2045116>. This way you can continue to use *Pythium sensu lato*, but distinguish between the different clades/species in the *Pythiales*.

Answer:

We thank the reviewer for highlighting the taxonomic history and development of *Pythium*. As suggested, we clarified this now with a short section in the introduction and hope that it meets the reviewer's requirement (lines 35-49).

In addition, the terminology *Pythium* s.l. is consequently used in the revised version, as recommended by the reviewer. Actually, we have recognized the problematics regarding *P. heterothallicum* and *G. heterothallicum*. But the costume database (Genbank) we used was not able to clearly distinguish them, whereby some species were annotated as *Pythium* and some as *Globisporangium*, depending on the best hits annotated. This problem has been now solved by using an updated version of the taxonomic classification of genera according to Nguyen et. al. 2022 (<https://doi.org/10.1080/00275514.2022.2045116>).

Furthermore, we integrated the different clades in Figure 2, but differing from Figure 3 of Rojas et al. (2017) (<http://dx.doi.org/10.1094/PHYTO-04-16-0177-R>). *Pythium* s.l. is now used as recommended and the different clades/species within the *Pythiales* were distinguished. We hope that this figure, although slightly different from the reviewer's suggestion, is still in line with the reviewer's interest.

Comment 3:

The number of ASVs associated with "*Pythium* sp." You indicate there are 77 ASVs of *Pythium* sp. Are you sure that all of these ASVs represent a single phylogenetically distinct *Pythium* species or do they represent multiple "*P. sp.*". This needs to be clarified in the methods. How do you assign taxonomy to unknowns? Based on your methods

you custom database would include all "Pythium sp" in Genbank, which may compose dozens of phylogenetically distinct species. There may be multiple phylogenetically distinct species in this group of 77 ASVs. Clarification is needed.

Answer:

No, the number of ASV associated with "Pythium sp." does not represent phylogenetically distinct species. We see that they only concern several species of "P. sp". This explains well why many different "P. sp" cannot be accurately assigned in this way in our previous version.

This problem has fortunately also been solved in the revised version by using the database recommended by the reviewer. In our opinion, further clarification in the materials and methods is therefore no longer necessary.

Comment 4:

All figures are draw into questions based on the dual use of Globisporangium and Pythium names for the same species. See notes in attached document.

Answer:

Thanks for the remark. We have revised all figure labels and legends in the revised version according to the results of the re-analysis as suggested by the reviewer.

Comment 5:

All uses of "Pythium" should be italicized unless you are referring to the disease such as "Pythium root rot" or "Pythium seedling disease".

Answer:

Many thanks! We have corrected it through the whole text.

Comment 6:

Additional comments and suggested edits are included in that attached document.

Answer:

Thanks for the efforts from the reviewer. We have revised the text accordingly.

To reviewers 2 and 3:

We would like to thank both reviewers for their great efforts and help to improve our manuscript.

We are very pleased about their positive evaluation to our revised manuscript.

Prof. Dr. Daguang Cai

Kiel, 07.06.2024

REVIEWERS' COMMENTS

Reviewer #1 (Remarks to the Author):

I want to thank the authors for taking the time to make the modifications recommended by this reviewer. I understand this added significant time and effort to the process, but I believe this paper will be widely read and cited. This will lead to many authors also replicating this work in other locations. It is therefore, quite important that accurate databases be used.

I am very please with the edits and modifications made. I have no further comments or edits.